# Imaging quantized vortex rings in superfluid helium to evaluate quantum dissipation

Yuan Tang[1,2], Wei Guo [1,2] ✉, Hiromichi Kobayashi [3,4], Satoshi Yui[5,6], Makoto Tsubota[5,6] & Toshiaki Kanai [1,7]

The motion of quantized vortices is responsible for many intriguing phenomena in diverse quantum-fluid systems. Having a theoretical model to reliably predict the vortex motion therefore promises a broad significance. But a grand challenge in developing such a model is to evaluate the dissipative force caused by thermal quasiparticles in the quantum fluids scattering off the vortex cores. Various models have been proposed, but it remains unclear which model describes reality due to the lack of comparative experimental data. Here we report a visualization study of quantized vortex rings propagating in superfluid helium. By examining how the vortex rings spontaneously decay, we provide decisive data to identify the model that best reproduces observations. This study helps to eliminate ambiguities about the dissipative force acting on vortices, which could have implications for research in various quantum-fluid systems that also involve similar forces, such as superfluid neutron stars and gravity-mapped holographic superfluids.

Many quantum fluids, such as superfluid helium-4 (He II), can be considered as a mixture of two miscible fluid components: an inviscid superfluid and a viscous normal fluid consisting of thermal quasiparticles[1]. A conspicuous feature of the superfluid is the existence of topological defects in the form of quantized vortices[2]. In 3D space, these vortices appear as density-depleted thin tubes, each carrying a circulating flow with a fixed circulation $\kappa = h/m$, where $h$ is Planck's constant and $m$ is the mass of the bosons constituting the superfluid[2]. The motion of quantized vortices is responsible for a wide range of phenomena in quantum-fluid systems, such as the emergence of quantum turbulence in He II and atomic Bose-Einstein condensates[3–5], the initiation of dissipation in type-II superconductors[6], the appearance of glitches in neutron star rotation[7,8], and the formation of possible cosmic-string network[9]. As the vortices move through the normal fluid, a mutual friction between the two fluids can arise due to the scattering of the thermal quasiparticles off the vortex cores[10–12]. Understanding the dynamics of quantized vortices in the presence of

the normal fluid is therefore of broad significance. However, despite decades of research[13–18], the lack of experimental data has left the issue of how to accurately model the mutual friction unsettled.

In the pioneering work of Schwarz[13,14], a vortex filament model was developed for studying turbulence in He II. In this model, the quantized vortices are described by zero-thickness filaments that are divided into small segments. A vortex segment with a length $\Delta \xi$ located at $\mathbf{s}$ would experience a Magnus force $\mathbf{f}_M = \rho_s \kappa \mathbf{s}' \times (\mathbf{u}_L - \mathbf{u}_s) \Delta \xi$ when its velocity $\mathbf{u}_L$ differs from the local superfluid velocity $\mathbf{u}_s$. Here $\mathbf{s}'$ is the unit tangent vector along the filament, and $\rho_s$ is the superfluid density. Besides, any relative motion between the vortex segment and the normal fluid can result in a mutual friction force as proposed by Schwarz $\mathbf{f}_{sn} = [-\gamma_0 \mathbf{s}' \times (\mathbf{s}' \times (\mathbf{u}_n - \mathbf{u}_L)) + \gamma_0' \mathbf{s}' \times (\mathbf{u}_n - \mathbf{u}_L)] \Delta \xi$, where $\gamma_0$ and $\gamma_0'$ are temperature-dependent empirical coefficients[14]. By balancing the two forces, Schwarz obtained the vortex equation of motion (see Methods), which has been extensively employed in past vortex research[19–22].

[1]National High Magnetic Field Laboratory, 1800 East Paul Dirac Drive, Tallahassee, FL 32310, USA. [2]Mechanical Engineering Department, FAMU-FSU College of Engineering, Florida State University, Tallahassee, FL 32310, USA. [3]Research and Education Center for Natural Sciences, Keio University, 4-1-1 Hiyoshi, Kohoku-ku, Yokohama 223-8521, Japan. [4]Department of Physics, Hiyoshi Campus, Keio University, 4-1-1 Hiyoshi, Kohoku-ku, Yokohama 223-8521, Japan. [5]Department of Physics, Osaka Metropolitan University, 3-3-138 Sugimoto, Sumiyoshi-ku, Osaka 558-8585, Japan. [6]Nambu Yoichiro Institute of Theoretical and Experimental Physics (NITEP), Osaka Metropolitan University, Osaka 558-8585, Japan. [7]Department of Physics, Florida State University, Tallahassee, FL 32306, USA. ✉ e-mail: wguo@magnet.fsu.edu

However, a known limitation of the Schwarz model is that the normal-fluid velocity $\mathbf{u}_n$ is prescribed and there is no back action from the vortices to the normal fluid. To fix this issue, a two-way (2W) model was later developed, where $\mathbf{u}_n$ is solved using the Navier-Stokes equation with an added mutual-friction term that couples to the vortices. This model has allowed researchers to explain puzzling observations in He II turbulence[18,23]. Nonetheless, it was postulated that the coefficients $\gamma_0$ and $\gamma_0'$ may not be applicable to individual vortices since they were deduced from measurements where $\mathbf{u}_n$ was averaged over an array of vortices[15]. Over the past two decades, researchers have strived to calculate the friction coefficients in a self-consistent manner[15-17]. These efforts led to the striking prediction of the triple-vortex-ring structure in He II[15,17,24]. Recently, Galantucci et al. derived the most refined version of the self-consistent two-way (S2W) model where the mutual friction coefficient can be calculated directly from $\mathbf{u}_n$ without relying on any empirical experimental input[17].

These different models render distinct normal-fluid flow structures around the quantized vortices, which affect the vortex motion. As an example, we show in Fig. 1a the calculated normal-fluid velocity field around a quantized vortex ring in quiescent He II using all three models (see Methods for details). Unlike the Schwarz model where $\mathbf{u}_n = 0$, both the 2W model and the S2W model reveal the existence of two oppositely polarized normal-fluid vortex rings sandwiching the quantized vortex ring. These normal-fluid rings affect the local $\mathbf{u}_n$ experienced by the quantized ring and hence can alter the mutual friction dissipation. However, is this triple-ring structure real? If so, which model better describes the true vortex dynamics? These questions are important but have remained open due to the lack of experimental information. In this work, we provide the long-awaited data to show that the S2W model can better reproduce experimental observations. These findings may have the potential to improve our comprehension of various vortex-involved phenomena in quantum two-fluid systems.

## Results

### Visualizing quantized vortex rings

To study the vortex motion, we visualize quantized vortices in He II by decorating them with solidified deuterium ($D_2$) tracer particles[25,26]. This method has already allowed researchers to gain valuable insights into the properties of tangled vortices[27-31]. In an earlier work by Bewley and Sreenivasan[32], they reported on the visualization of a single vortex ring event. However, as these authors pointed out, many tracers had condensed onto the vortex core, which could alter the core size and hence the ring dynamics. The low vortex-line density in that experiment also limited the chance to observe more vortex rings. Based on the insights gained from this valuable work, we have designed our experimental setup to facilitate improved vortex-ring imaging. As shown in Fig. 1b, we control the vortex-line generation by towing a mesh grid in a plexiglass channel ($1.6 \times 1.6 \times 33\ cm^3$) immersed in a He II bath. The bath temperature can be controlled with an accuracy of 1 mK by regulating the vapor pressure. Following the grid motion, a mixture of $D_2$ gas and $^4$He gas is injected into the channel at about 30 s delay so that the background flow is weak but vortices with a line density of the order $10^2\ cm^{-2}$ still remain[33,34]. The higher line density can increase the likelihood of observing vortex rings, since the reconnections of intersecting vortex lines in the tangle can give rise to them. The $D_2$ gas forms ice particles (density[35]: $\rho_p = 202.8\ g/cm^3$) with a mean radius of 1.1 μm as determined from their settling velocities (see Methods and Supplementary Fig. 1). When the $D_2$ particles are close to the vortex cores, they get trapped on the vortices due to a Bernoulli pressure caused by the circulating superfluid[36]. Through extensive trials, we have figured out the optimal injection conditions to achieve the desired particle number density on the vortices. The particles are then illuminated by a laser sheet (thickness 0.8 mm, see Supplementary Fig. 4) and their positions are recorded at 200 Hz by a video camera placed perpendicular to the laser plane. From time to time, we can see vortex rings propagating within the laser sheet. A collection of representative ring events obtained at different temperatures is included in

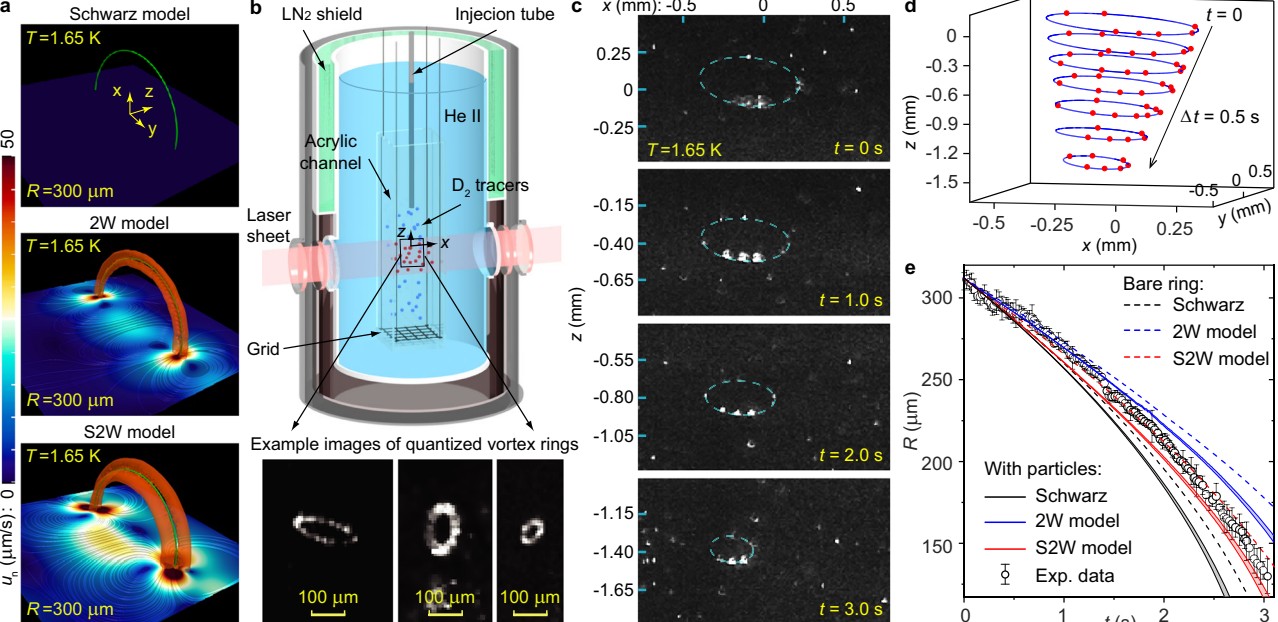

**Fig. 1 | Modeling and imaging quantized vortex rings in He II. a** Calculated normal-fluid velocity field $\mathbf{u}_n$ around a quantized vortex ring in He II. Due to the axial symmetry, we only show $\mathbf{u}_n$ in the $y$-$z$ plane and the vortex ring above the plane (i.e., the green curve). The normal-fluid vortex rings (reddish half circles) are rendered in the same way as in ref. 18. **b** Schematic diagram of the experimental setup. **c** Images showing the $D_2$ particles (white dots) trapped on a moving vortex ring in quiescent He II. The dashed ellipse is a fit to the trapped particles' positions. **d** Obtained vortex-ring profile with the trapped particles (red dots) at different times. **e** Comparison of the observed ring radius $R(t)$ evolution with model predictions. The error bars on the $R(t)$ data denote the standard deviation derived from the positional uncertainties of the particles in the ellipse fit. The narrow color-shaded areas depict the variation range of the simulation curves for the particle-doped ring, as the radius $a$ of each trapped particle is adjusted from $a - \Delta a$ to $a + \Delta a$.

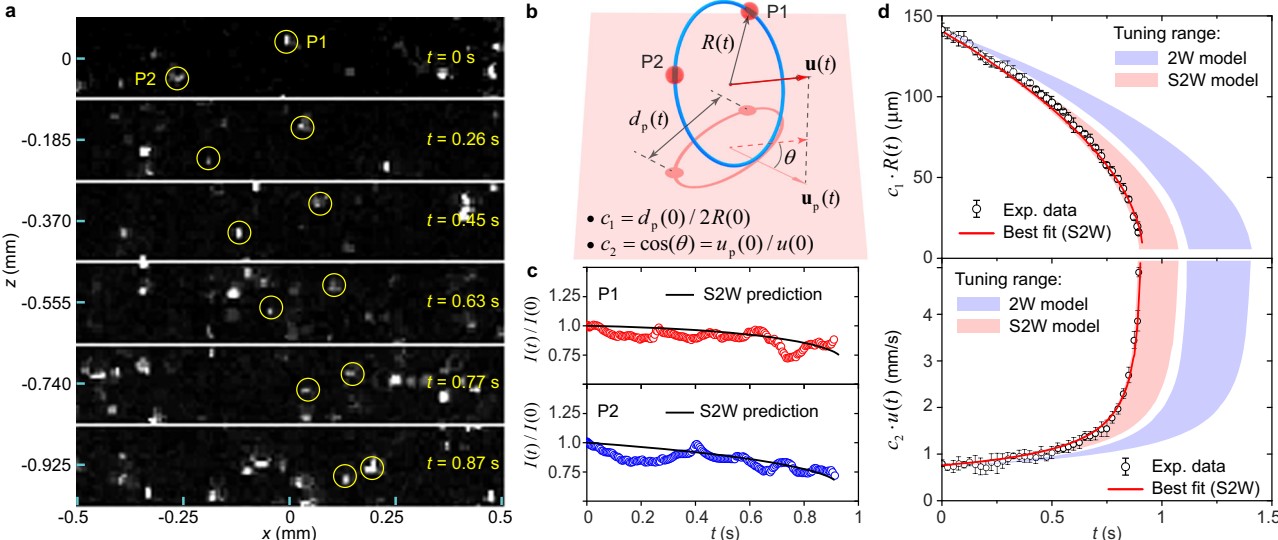

**Fig. 2 | Analysis of a vortex ring with two trapped particles. a** Images showing two trapped particles (circled) moving in He II at 1.65 K. **b** A schematic explaining the concept of the projection parameters $c_1$ and $c_2$. **c** Variation of the corrected brightness $I(t)$ of the two trapped particles. The solid curve shows the prediction of the S2W model that gives the best fit to the ring's radius and velocity data in **d**.

**d** Comparison of the projected ring radius and velocity data with model simulations. The tuning range of the simulation curve is set by the constraints on $R(0)$ and $u(0)$, as discussed in the text. The error bars denote the standard deviation associated with the data due to the position uncertainties of the two trapped particles (see Methods).

Supplementary Movie 1. We have also captured videos showing how vortex rings are created by vortex-line reconnections (see Supplementary Movie 2). Note that as temperature rises, the lifetime of vortex rings reduces due to increased dissipation, which ultimately decreases the likelihood of capturing high-quality ring events. In the following sections, we present our best data obtained at 1.65 K for comparison with model simulations.

### Data analysis and model comparison

To extract useful information on vortex-ring propagation, we focus on analyzing selected events where the rings are decorated by discrete $D_2$ particles and move in He II with negligible background flows. A good example is shown in Fig. 1c where the ring moves downward carrying nine $D_2$ particles (see Supplementary Movie 3). We first use a feature-point tracking routine[37] to determine the positions of the trapped particles in each image. Then, the particle positions are fitted with an ellipse. This fitting, which requires at least 5 particles on the ring, allows us to determine both the ring radius $R$ and the orientation of the ring plane (see Methods). Figure 1d shows the extracted ring profile with the trapped particles at different times. The ring shrinks due to the mutual friction dissipation, which leads to an acceleration of its self-induced motion[36]. By analyzing the angular positions of the particles on the ring over time, we confirm that these particles exhibit little movement along the circumference of the ring (see Methods and Supplementary Fig. 2), a puzzle that has been actively discussed in literature[38]. In Fig. 1e, we show the obtained ring radius $R(t)$. These data are shown at every five image frames instead of at 200 Hz for better visibility of the error bars. For comparison, we also include the simulated $R(t)$ for a bare vortex ring in quiescent He II with the same initial radius using all three models. The result suggests that the S2W model renders the best agreement with the data.

However, it is well known that the trapped particles can result in additional forces on the vortex core[39–41], which needs to be considered when simulating the ring's motion. Following Mineda et al.[39] (see Methods), we consider the Stokes drag[1] $\mathbf{f}_D = -6\pi a\mu_n(\mathbf{u}_L - \mathbf{u}_n)$, the gravitational force, and the inertial effect of each trapped particle on the ring. Here $\mu_n$ is the He II dynamic viscosity and $a$ is the particle radius. To evaluate $a$, we first develop a correlation between the particle's brightness and its radius by comparing the distributions of these

two quantities (see Methods and Supplementary Fig. 3). We then examine the time-averaged brightness of each trapped particle and calculate its radius using the correlation. The obtained radiuses are listed in the Supplementary Table 1. With this information, we can re-calculated $R(t)$ using the three models (see Fig. 1e). Due to the additional Stokes drag, the ring shrinks faster in all three models. Obviously, the Schwarz model overestimates the dissipation and can be rejected. But it becomes less clear whether the S2W model still describes the data better than the 2W model. To make a reliable assessment on these two models, it is imperative to analyze rings with a minimal number of trapped particles, since the uncertainties in the particle positions and sizes could affect both the $R(t)$ data and the simulation curves.

Luckily, we have recorded several unique events where the rings are decorated by only two $D_2$ particles (see Supplementary Movie 4). For these events, the estimated Stokes drag and the gravitational force are only a few percent of the mutual friction (see Methods). Figure 2a shows our best example, where two particles $P_1$ and $P_2$ move in sync while approaching each other due to the shrinkage of the vortex ring. We can measure the separation distance $d_p(t)$ between the two particles and their centroid velocity $u_p(t) = |\frac{1}{2}(\mathbf{u}_1 + \mathbf{u}_2)|$. However, as illustrated in Fig. 2b, $d_p(t)$ in general does not equal the vortex-ring diameter $2R(t)$, and $u_p(t)$ can differ from the actual ring velocity $u(t)$ since a projection angle $\theta$ may exist between the ring's propagation direction and the laser plane. In order to utilize the experimental data for model comparison, we adopt the following procedures. First, we assume an initial ring radius $R(0)$ and calculate the evolution of the ring's radius $R(t)$ and velocity $u(t)$ using both the 2W and the S2W models. Next, we evaluate two projection parameters $c_1 = d_p(0)/2R(0)$ and $c_2 = \cos(\theta) = u_p(0)/u(0)$. These two parameters remain nearly constant because: 1) the particles do not slide along the vortex core as we learned from the study of rings with 5 or more trapped particles; and 2) the centroid of $P_1$ and $P_2$ moves in a straight path, suggesting a constant projection angle. Using $c_1$ and $c_2$, we can then compare $c_1R(t)$ and $c_2u(t)$ directly with the experimental data $d_p(t)/2$ and $u_p(t)$. Finally, we vary $R(0)$ to see which model can render results that simultaneously match $d_p(t)/2$ and $u_p(t)$.

In this analysis, there are a few constraints on the range of $R(0)$ that we can explore. First, $R(0) \geq d_p(0)/2$ since the two particles cannot

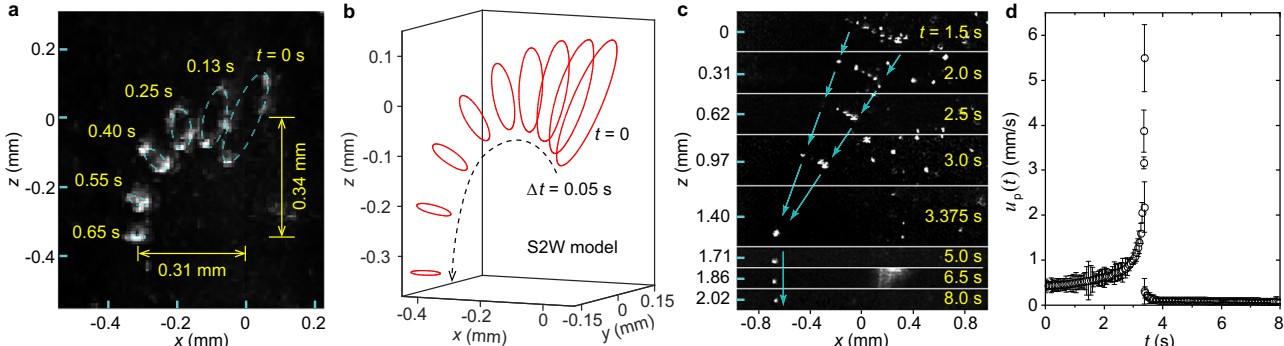

**Fig. 3 | Other intriguing observations of the vortex rings. a** A superimposed image showing a heavily doped vortex ring gradually flipping downward while it shrinks. **b** Simulated motion of a vortex ring with the same initial profile carrying 36 $D_2$ particles (4.9 μm in radius) using the S2W model. **c** Images showing how the $D_2$ particles trapped on a vortex ring eventually form a cluster that falls freely in He II. **d** Measured centroid velocity of the trapped $D_2$ particles shown in **c**. The error bars denote the standard deviation associated with the data due to the uncertainties of the centroid position (see Methods).

be separated by more than the diameter of the ring. Second, $u(0) \geq u_p(0)$ due to the projection, which sets an upper limit of $R(0)$ because $u(0)$ drops as $R(0)$ increases. The last constraint comes from the observed particle brightness $I$ after correcting the $z$-dependance of the laser intensity (see Methods). As shown in Fig. 2c, $I$ for either particles only drops by about 20% during the ring's propagation. Based on the cross-sectional profile of the laser sheet (see Supplementary Fig. 4), we estimate that the ring can move by at most 0.2 mm perpendicular to the laser plane. This sets an upper limit of the projection angle $\theta$, which constrains $u(0)$ and hence $R(0)$. In Fig. 2d, we show the calculated $c_1 R(t)$ and $c_2 u(t)$ using the 2W and the S2W models while $R(0)$ is varied in the range set by all the constraints. Clearly, the experimental data are outside the tuning range of the 2W model. On the other hand, we find that the S2W model can nicely reproduce both $d_p(t)/2$ and $u_p(t)$ data at $R(0) = 140.8$ μm. This optimal $R(0)$ happens to be close to $d_p(0)/2 = 140.6$ μm, suggesting that the two particles were located nearly across the diameter of the vortex ring. In conclusion, our 2-particle ring analysis indicates that the S2W model has better fidelity than the 2W model.

### Other intriguing observations

Besides model testing, we would also like to report some intriguing phenomena that have not been reported in literature. The first phenomenon is that sometimes we see vortex rings heavily doped with $D_2$ particles can spontaneously flip to the downward direction. A collection of such events is included in Supplementary Movie 5. In Fig. 3a, we superimpose the images of a representative ring taken at different $t$ to show how the ring changes its direction while it shrinks. This phenomenon can be understood by noting that the vortex ring carries a momentum[36] $\mathbf{P}(t) = \rho_s \kappa \pi R(t)^2 \hat{\mathbf{n}}$, where $\hat{\mathbf{n}}$ is the unit vector normal to the ring plane pointing in the direction of the ring's motion. The mutual friction and the Stokes drag constantly reduce the ring's momentum, resulting in the shrinkage of the ring. On the other hand, the gravitational force from the trapped particles continuously generates momentum in the downward direction, which forces the ring to flip downward. To test this understanding, we have conducted simulations using the S2W model. For a heavily doped ring, the exact number $N$ and the radiuses of the trapped particles are hard to determine. Instead, we assume the same radius $a$ for all the trapped particles and adjust both $a$ and $N$ in our trials. For the ring trajectory presented in Fig. 3a, we find that it can be qualitatively reproduced with $N = 36$ and $a = 4.9$ μm, as shown in Fig. 3b. Similar flipping can be also achieved with fewer but heavier trapped particles.

Another intriguing observation is related to the destiny of the particles on the vortex rings. As a ring shrinks, we often see that the trapped particles form a cluster whose high-speed motion changes abruptly to slowly falling in He II (see Supplementary Movie 6). Figure 3c shows an event where the ring plane is nearly perpendicular to the laser plane. Nonetheless, we can measure the centroid velocity $u_p(t)$ of the particles. As shown in Fig. 3d, $u_p(t)$ increases drastically as the ring shrinks. At $t = 3.375$ s, the trapped particles aggregate to a single cluster and $u_p(t)$ suddenly drops to the expected settling velocity of about 0.1 mm/s. Our interpretation of this phenomenon is that as the ring shrinks, its velocity relative to the normal fluid becomes so large such that the Stokes drag can pull the trapped particles off the vortex core. Subsequently, the bare ring moves away and diminishes, while the left-behind particles form a cluster that decelerates rapidly to the settling velocity due to the Stokes drag. To test this hypothesis, we estimate the maximum trapping force on a particle from the vortex core as[42] $f_v \simeq \rho_s \kappa^2/3\pi$ and compare it with the Stokes drag $f_D \simeq 6 \pi a \mu_n u_p$. For particles with a mean radius $a \simeq 1$ μm, $f_D$ becomes greater than $f_v$ when $u_p$ reaches a threshold value of 5.1 mm/s. This threshold $u_p$ is close to the observed maximum $u_p$ in Fig. 3d, which provides a strong support to our interpretation.

### Discussion

The results that we have presented provide the first evidence suggesting that the S2W model is more consistent with observed vortex dynamics in He II in comparison to alternative models. This study may stimulate future research in two possible directions. First, the S2W model does not rely on empirical experimental inputs and therefore can be readily adapted for other quantum two-fluid systems, such as BECs[43,44], superfluid neutron stars[7,8,45], and gravity-mapped holographic superfluid[46,47]. An accurate evaluation of the mutual friction is particularly important for processes that involve rapid motion of the quantized vortices, such as vortex reconnections, and pinning and depinning of vortices on solid boundaries. The latter process is the key for understanding glitches in neutron star rotation[45]. The second direction is to examine how the implementation of the S2W model may alter our existing knowledge on quantum turbulence (QT) induced by a chaotic tangle of quantized vortices. For instance, an important topic in QT research is counterflow turbulence where the mutual friction exists at all length scales[48,49]. Our knowledge on the vortex-tangle properties[14,19], disturbances in the normal fluid[18,50], and the effect of the mutual friction on the mean-velocity profile[23,51,52] may subject to change with future S2W-model simulations.

### Methods

#### Numerical models

**Schwarz model.** In the framework of Schwarz's vortex filament model[14], all the quantized vortex lines are represented by zero-thickness filaments. The position vector of a filament can be written in

the parametric form $\mathbf{s} = \mathbf{s}(\xi, t)$, where $\xi$ denotes the arc length along the filament. In the presence of the viscous normal fluid, a short segment $\Delta\xi$ of a vortex filament located at $\mathbf{s}$ would experience two forces, i.e., the Magnus force $\mathbf{f}_M = \rho_s \kappa \mathbf{s}' \times (\mathbf{u}_L - \mathbf{u}_s)\Delta\xi$ and the mutual friction force $\mathbf{f}_{sn} = [-\gamma_0 \mathbf{s}' \times (\mathbf{s}' \times (\mathbf{u}_n - \mathbf{u}_L)) + \gamma_0' \mathbf{s}' \times (\mathbf{u}_n - \mathbf{u}_L)]\Delta\xi$. By balancing these two forces, the velocity of this segment $\mathbf{u}_L = d\mathbf{s}/dt$ can be derived as:

$$d\mathbf{s}/dt = \mathbf{u}_s + \alpha \mathbf{s}' \times (\mathbf{u}_n - \mathbf{u}_s) - \alpha' \mathbf{s}' \times [\mathbf{s}' \times (\mathbf{u}_n - \mathbf{u}_s)], \qquad (1)$$

where the coefficients $\alpha$ and $\alpha'$ depend on the empirical mutual friction coefficients $\gamma_0$ and $\gamma_0'$, whose values have been tabulated[36]. While the normal-fluid velocity $\mathbf{u}_n$ is prescribed, the local superfluid velocity $\mathbf{u}_s(\mathbf{s}, t)$ is evaluated as the sum of the background flow velocity $\mathbf{u}_{s0}$ and the velocity $\mathbf{u}_{in}$ induced at $\mathbf{s}$ by all the vortices, which can be calculated using the full Boit-Savart integral[19]:

$$\mathbf{u}_{in}(\mathbf{s}, t) = \frac{\kappa}{4\pi} \int \frac{(\mathbf{s_1} - \mathbf{s}) \times d\mathbf{s_1}}{|\mathbf{s_1} - \mathbf{s}|^3}, \qquad (2)$$

where the integration goes over all the vortex filaments. When we apply the Schwarz model to simulate the motion of a vortex ring in quiescent He II, we set both $\mathbf{u}_n$ and $\mathbf{u}_{s0}$ to zero and discretize the initial ring with a resolution $\Delta\xi = 0.005$ mm. The time evolution of each vortex segment's position can be obtained through a temporal integration of Eq. (1) using the fourth-order Runge-Kutta method[53] with a time step $\Delta t = 10^{-5}$ s.

**2W model.** In the 2W model, the normal-fluid velocity $\mathbf{u}_n$ is no longer prescribed. Instead, it is calculated by solving the classical Navier-Stokes equation with an added mutual friction term[18]:

$$\frac{\partial \mathbf{u}_n}{\partial t} + (\mathbf{u}_n \cdot \boldsymbol{\nabla})\mathbf{u}_n = -\frac{1}{\rho_{He}}\boldsymbol{\nabla}P + \nu_n \nabla^2 \mathbf{u}_n + \frac{\mathbf{F}_{ns}}{\rho_n} \qquad (3)$$

where $\rho_n$ and $\rho_{He}$ are, respectively, the normal-fluid density and the total density of He II, $P$ is the pressure, $\nu_n$ is the He II kinematic viscosity, and $\mathbf{F}_{ns}$ is the mutual friction per unit volume which can be calculated as:

$$\mathbf{F}_{ns}(\mathbf{r}) = \frac{1}{\Delta\Omega(\mathbf{r})} \int_{\mathcal{L}(\mathbf{r})} (-\mathbf{f}_{sn}/\Delta\xi)d\xi \qquad (4)$$

where $\mathcal{L}(\mathbf{r})$ denotes that the integration is performed along all the vortex lines in the computational cell $\Delta\Omega(\mathbf{r}) = \Delta x \times \Delta y \times \Delta z$ located at $\mathbf{r}$. When we simulate the vortex ring dynamics, Eqs. (1) and (3) are solved together to render the positions of the vortex-ring segments $\mathbf{s}(\xi, t)$ and the normal-fluid velocity $\mathbf{u}_n$. The time integration of Eq. (3) is conducted using the second-order Adams-Bashforth method[18] with the same time step $\Delta t = 10^{-5}$ s, and the spatial differentiation is performed via the second-order finite difference with a spatial resolution $\Delta x = \Delta y = \Delta z = 0.0083$ mm. The computational domain consists of $120^3$ grids. We have confirmed that finer spatial and temporal resolutions do not change the simulation results.

**S2W model.** In the S2W model, the mutual friction force that acts on a vortex segment $\Delta\xi$ is given by[17]:

$$\mathbf{f}_{sn} = [-D\mathbf{s}' \times (\mathbf{s}' \times (\mathbf{u}_n - \mathbf{u}_L)) - \rho_n \kappa \mathbf{s}' \times (\mathbf{u}_n - \mathbf{u}_L)]\Delta\xi, \qquad (5)$$

where the only friction coefficient $D$ can be calculated as:

$$D = -4\pi\rho_n\nu_n/[0.0772 + \ln(|\mathbf{u}_n^{\perp} - \mathbf{u}_L|a_0/4\nu_n)]. \qquad (6)$$

Here $a_0 \simeq 1$ Å is the vortex-core radius[36] and $\mathbf{u}_n^{\perp}$ denotes the local normal-fluid velocity at the vortex-segment location that is projected

in the plane perpendicular to the segment[17]. By balancing the Magnus force $\mathbf{f}_M$ and the revised mutual friction force, the equation of motion for the vortex segment is now given by:

$$d\mathbf{s}/dt = \mathbf{u}_s + \beta \mathbf{s}' \times (\mathbf{u}_n - \mathbf{u}_s) - \beta' \mathbf{s}' \times [\mathbf{s}' \times (\mathbf{u}_n - \mathbf{u}_s)], \qquad (7)$$

where the coefficients $\beta$ and $\beta'$ depends on $D$ as derived by Galantucci et al.[17]. The evolution of the vortex position and $\mathbf{u}_n$ can be obtained by solving Eqs. (3) and (7) with $D$ evaluated self-consistently via Eq. (6). In the S2W simulations, we adopted the same temporal and spatial resolutions as for the 2W simulations.

Note that the hydrodynamic description of the normal fluid is applicable only when the roton mean free path in He II is much smaller than the relevant length scales. This path is about 30 nm at 1 K and decreases with increasing the temperature[54]. Therefore, as we study a vortex ring of tens to hundreds of microns in diameter at temperatures above 1.6 K, the hydrodynamic description of the normal fluid should be reasonable.

For a quantized vortex ring with a radius $R$ moving in quiescent He II, the self-induced superfluid velocity at the ring's location is given by[36] $\mathbf{u}_s = \frac{\kappa}{4\pi R}[\ln(8R/a_0) - \frac{1}{2}]\hat{\mathbf{n}}$, which is the same in all three models. However, the local $\mathbf{u}_n$ is different, which leads to the different mutual friction dissipation rate. In the Schwarz model, $\mathbf{u}_n = 0$ and therefore the highest mutual friction dissipation is expected. In both the 2W model and the S2W model, the back action of the mutual friction in the normal fluid generates two oppositely polarized normal-fluid vortex rings as shown in Fig. 1. In the 2W model, the two normal-fluid rings are concentrically located nearly in the same plane as the quantized vortex ring, whereas in the S2W model the two normal-fluid rings are slightly shifted to above and below the quantized-ring plane. This shift changes the direction of the local $\mathbf{u}_n$. Nonetheless, the induced local $\mathbf{u}_n$ in both models has a component in the same direction as the local $\mathbf{u}_s$, which effectively reduces the mutual friction dissipation as compared to that in the Schwarz model.

### Effects of the trapped particles
When a vortex segment $\Delta\xi$ carries a trapped particle with a radius $a$, its equation of motion changes to[39]:

$$(m_p + m_f)\frac{d\mathbf{u}_L}{dt} = \mathbf{f}_M + \mathbf{f}_{sn} + \mathbf{f}_D + \mathbf{f}_g \qquad (8)$$

where the term on the left-hand side represents the inertial effect caused by the trapped particle's mass $m_p = \rho_p \frac{4}{3}\pi a^3$ and the fluid's added mass $m_f = \frac{1}{2}\rho_{He}\frac{4}{3}\pi a^3$. On the right-hand side, besides the Magnus force $\mathbf{f}_M$ and the mutual friction force $\mathbf{f}_{sn}$, two additional forces are included, i.e., the Stokes drag exerted by the normal fluid on the particle $\mathbf{f}_D = -6\pi a\mu_n(\mathbf{u}_L - \mathbf{u}_n)$ and the gravitational force $\mathbf{f}_g = (\rho_p - \rho_{He})\frac{4}{3}\pi a^3 \mathbf{g}$. Other minor effects associated with the acceleration of the superfluid and the normal fluid around the trapped particle are negligible[39]. This model is accurate when $a$ is much smaller than the separation distance between the particles trapped along the vortex ring, which holds true for the ring events that we selected to analyze.

To get a sense on how large the particle effects are, one may compare the total Stokes drag $F_D = |\sum_i \mathbf{f}_{D,i}|$ and the total gravitational force $F_g = |\sum_i \mathbf{f}_{g,i}|$ with the total mutual friction force $F_{sn} = |\oint(\mathbf{f}_{sn}/\Delta\xi)d\xi|$, where $\sum_i$ means the summation over all the trapped particles and $\oint$ denotes the integration along the vortex ring. For the 9-particle vortex-ring event shown in Fig. 1, using the particle radiuses obtained through the size analysis (see later discussions in Methods), we estimate that $F_D/F_{sn}$ and $F_g/F_{sn}$ vary in the range of 10−18% and 4−4.7%, respectively, as $R(t)$ decreases from 312 μm at $t = 0$ to 150 μm. On the other hand, for the 2-particle vortex-ring event shown in Fig. 2, despite the ring's smaller initial radius and hence higher propagation speed, the

ratios are $F_D/F_{sn} \simeq 4.8 - 10\%$ and $F_g/F_{sn} \simeq 0.8 - 0.83\%$, respectively, as $R(t)$ shrinks from 140.8 μm at $t = 0$ to 50 μm.

## Particle size distribution

We produce solidified $D_2$ tracer particles in He II by slowly injecting a mixture of 5% $D_2$ gas and 95% $^4$He gas directly into the plexiglass channel immersed in the He II bath. A computer-controlled solenoid valve is used to adjust the injection duration, and a needle valve is adopted to restrict the gas flow rate. Upon the injection, the $D_2$ gas forms solidified ice particles. To evaluate the sizes of the resulted particles, we took images of the particles undergoing freely settling in quiescent He II in each experimental run. As a representative video, Supplementary Movie 7 displays the settling behavior of particles during the experiment run where we recorded the 9-particle ring event and the 2-particle ring event depicted in Fig. 1 and Fig. 2, respectively. By tracking the particles in such videos, we can generate a probability distribution of the particle settling velocity $u_p^{(s)}$. The result is shown in the Supplementary Fig. 1a. The $u_p^{(s)}$ data can be fitted nicely with a lognormal distribution, from which we can determine that the distribution is peaked at about 0.1 mm/s.

Note that the settling velocity is achieved when the Stokes drag exerting on a $D_2$ particle is balanced by the gravitational force, i.e., $6\pi a \mu_n u_p^{(s)} = \frac{4\pi}{3} a^3 (\rho_p - \rho_{He}) g$. This balance leads to $a = [9\mu_n u_p^{(s)}/2(\rho_p - \rho_{He})g]^{1/2}$. Therefore, knowing the distribution of $u_p^{(s)}$, we can then generate the radius distribution of the $D_2$ particles. As shown in Supplementary Fig. 1b, this distribution is peaked at $a \simeq 1.1$ μm with a root mean variance $\Delta a$ of about 0.2 μm.

## Positions and radiuses of trapped particles

To evaluate the effects of the trapped $D_2$ particles on the motion of a vortex ring, we need to know the radius and initial position of each individual trapped particle. Using the feature-point tracking routine[37], we can determine the coordinates of every particles in the $x$-$z$ image plane. For particles trapped on the vortex ring, their coordinates $(x_i, z_i)$ should satisfy the following equation of an ellipse:

$$\frac{[(x_i - x_0)\cos\phi + (z_i - z_0)\sin\phi]^2}{R_1^2} + \frac{[(z_i - z_0)\cos\phi - (x_i - x_0)\sin\phi]^2}{R_2^2} = 1,$$

(9)

where $(x_0, z_0)$ are the coordinates of the ellipse center, $R_1$ and $R_2$ are, respectively, the semi-major and semi-minor axes of the ellipse, and $\phi$ is the angle between the ellipse major axis and the $x$-axis. These five parameters can be uniquely determined through a least squares fit to the positions of the trapped particles when there are at least five particles on the ring. Through this fit, we can determine the vortex ring radius $R = R_1$ and the projection angle $\theta$ between the ring's normal vector $\hat{\mathbf{n}}$ and the $x$-$z$ plane (i.e., $\sin\theta = R_2/R_1$). If we set $y_0 = 0$ for the ellipse center at $t = 0$, the initial $y_i$ of each trapped particle can be calculated as $y_i = [(x_i - x_0)\sin\phi - (z_i - z_0)\cos\phi]/\tan\theta$. In the Supplementary Table 1, we list the 3D coordinates of all the nine trapped particles for the vortex ring presented in Fig. 1. The coordinate uncertainty comes from the feature-point tracking fit of the particle's image profile. These coordinates are used in our model simulations.

Knowing the coordinates of these trapped particles, we can also monitor their angular position on the vortex ring over time. We set the ellipse major axis as our reference and denote $\alpha$ as the angle between the position vector $\mathbf{OP}$ of a trapped particle with this reference axis, where $O$ is the center of the ring and $P$ denotes the position of the particle. In Supplementary Fig. 2, we show the time variation of $\alpha$ for a few representative particles of the 9-particle ring event. It is clear that these particles do not exhibit significant displacements along the ring.

To evaluate the trapped particle's radius $a$, we develop a correlation between $a$ and the particle's brightness $I$. For the particles that

undergo freely settling in Supplementary Movie 7, we calculate the brightness $I$ of each particle by summing up the counts in the image pixels associated with the particle. A distribution of the particle brightness $P(I)$ can therefore be generated, which is shown in Supplementary Fig. 3. Since $I$ depends on the particle's surface area and hence $a^2$, we can construct a simple correlation $I = A(a^2)^B$, where $A$ and $B$ are tuning parameters. For a given pair $A$ and $B$, we can scale the distribution of $a$ shown in the Supplementary Fig. 1b to generate the distribution of the expected brightness $I^{(ex)} = A(a^2)^B$. We then vary $A$ and $B$ to minimize the difference between the $I^{(ex)}$ distribution and the actual distribution $P(I)$. At the optimal values $A^* = 1.20$ and $B^* = 1.17$, the generated $I^{(ex)}$ distribution agrees nicely with $P(I)$, as shown in Supplementary Fig. 3.

Using the derived correlation $I = A^*(a^2)^{B^*}$, we can calculate the radius $a_i$ of a trapped particle $i$ by measuring its brightness $I_i$. However, we must note that this correlation holds only in a statistical sense. When we apply it to analyze the radiuses of individual particles, there can be intrinsic uncertainties. To improve the reliability, in practice we collect the brightness data of the particle $i$ over the time period that it is clearly observed and then use the time-averaged brightness $\bar{I}_i$ in the correlation to calculate $a_i$. The uncertainty of the particle radius $\Delta a_i$ can be evaluated as the variation range of $a_i$ when its brightness is varied from $\bar{I}_i - \Delta I_i$ to $\bar{I}_i + \Delta I_i$. Both the mean radiuses and the uncertainties are included in the Supplementary Table 1.

## Velocity and projection angle for 2P ring

For the 2-particle vortex ring event presented in Fig. 2, the centroid coordinates of the two particles can be easily caculated $\mathbf{x}_p = \frac{1}{2}(\mathbf{x}_1 + \mathbf{x}_2)$ and $\mathbf{z}_p = \frac{1}{2}(\mathbf{z}_1 + \mathbf{z}_2)$. To determine the centroid velocity $u_p$ at time $t$, we use a method that involves calculating the slopes of linear fits to the $x_p(t)$ and $z_p(t)$ data collected over a span of 5 consecutive image frames. Specifically, we determine the $x$-component of $u_p$ from the linear fit of the $x_p(t)$ data, and the $z$-component from the linear fit of the $z_p(t)$ data. The time range for data collection is from $t - 2\Delta t$ to $t + 2\Delta t$. This approach is consistently applied to all velocity calculations.

A constraint on the projection angle $\theta$ between the ring's propagation direction and the $x$-$z$ image plane can be placed based on the time-variation of the particle's brightness $I(t)$. This is because $\tan\theta = \Delta y / \Delta S$, where $\Delta S = 1.12$ mm is the distance traversed by the centroid of the two particles in the $x$-$z$ plane over the observation time $t_f = 0.89$ s, and $\Delta y$ is the centroid displacement in the $y$ direction perpendicular to the laser sheet, which can be estimated based on the variation of $I(t)$.

To estimate $\Delta y$, we first show the measured brightness $I_m(t)$ of each particle in the Supplementary Fig. 4a. The variation of $I_m(t)$ is caused by the displacement of the particles in both the $y$ direction and the $z$ direction, since $I_m(t)$ is proportional to the laser intensity $W$ which varies primarily in these two directions. To quantify the laser-intensity variations, we then place an optical power meter behind a mask with a narrow slit (20 μm in width) oriented either horizontally or vertically. By moving the horizontal slit in the $z$ direction or by moving the vertical slit in the $y$ direction, we can measure $W$ as a function of $y$ and $z$. The results are shown in the Supplementary Fig. 4b and c, respectively. The profile of $W$ in each direction can be reasonably fit with a Gaussian function, which renders $W(y, z) \propto e^{-2(y - y_c)^2/\sigma_y^2} \cdot e^{-2(z - z_c)^2/\sigma_z^2}$, where $y_c = 0$ and $z_c = -2.2$ mm are the coordinates of the beam's cross-sectional center, $\sigma_y = 0.69$ mm is the half-thickness of the laser sheet at $1/e^2$ intensity (i.e., which corresponds to a full thickness at half maximum intensity of 0.82 mm), and $\sigma_z = 3.5$ mm is the sheet's half-height at $1/e^2$ intensity.

Finally, we can calculate the corrected brightness $I(t) = I_m(t)/e^{-2(z(t) - z_c)^2/\sigma_z^2}$. The results are shown in Fig. 2c. The variation of $I(t)$ is entirely due to the particle displacement in the $y$ direction. Since $I(t)/I(0)$ for either particle decreases roughly monotonically by about 20% over the observation time, we can estimate the displacement $\Delta y$ based

on the Supplementary Fig. 4c. For a given initial particle coordinate $y(0)$, we can determine $\Delta y$ that gives 20% laser-intensity drop. By varying $y(0)$, we find that $\Delta y$ can reach up to about 0.2 mm. This sets an upper limit $\tan\theta \leq 0.2/1.12 = 0.18$. Since $\cos\theta = u_p(0)/u(0)$, a constraint on $u(0)$ and hence the initial ring radius $R(0)$ can be placed. This constraint together with the other constraints discussed in the paper render the tuning range of the simulation curves shown in Fig. 2d.

## Data availability

The data supporting the findings of this study are available within the paper and the Supplementary Information. Source data are also provided with this paper. Additional data related to this study are available from the corresponding author upon request. Source data are provided with this paper.

## Code availability

All computer codes used in this study are available from the corresponding author upon request.

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

## Acknowledgements

The authors would like to thank C. F. Barenghi for valuable discussions. Y.T., W.G., and T.K. are supported by the National Science Foundation under Grant No. DMR-2100790 and the Gordon and Betty Moore Foundation through Grant GBMF11567. They acknowledge the support and resources provided by the National High Magnetic Field Laboratory at Florida State University, which is supported by the National Science Foundation Cooperative Agreement No. DMR-2128556 and the state of Florida. Y.T., W.G., and T.K. would also like to acknowledge the use of the services provided by Florida State University Research Computing Center. M. T. acknowledges the support by the JSPS KAKENHI program under Grant No. JP20H01855. H.K. acknowledges the support by the JSPS KAKENHI program under Grant No. JP22H01403.

## Author contributions

W.G. designed and supervised the research and wrote the paper; Y.T. conducted the experiment; H.K. and Y.T. performed the numerical simulations; Y.T., W.G., H.K., S.Y., M.T., and T.K. participated in the result analysis and paper revision.

## Competing interests

The authors declare no competing interests.
