## [Peer Review File · Nature Communications]

Imaging quantized vortex rings in superfluid helium to evaluate quantum dissipationREVIEWER COMMENTS

Reviewer #1 (Remarks to the Author):

Attached file "Report.pdf"

Report

The authors of the paper “Imaging quantized vortex rings in superfluid helium to decipher quantum dissipation” present some experimental results on superfluid helium by using visualization techniques. They use solidified deuterium tracer particles for decorating quantized vortices in order to study the dynamics of vortex rings and hence to analyze the friction forces between quantized vortices and the quasiparticles composing the normal fluid component in the two-fluid model.

In their studies they are in particular interested to compare their results with the theoretical ones coming from three existing mathematical models, which describe the motion of the quantized vortices in superfluid helium taking into account the back-reaction of the quasi-particles (normal component) to the vortices through the mutual friction force. According to their studies they conclude that the best model is that proposed by Galantucci et al. in *Eur. Phys. J. Plus* 135, 547 (2020).

It is surprising to me that the mathematical model proposed by Mongiovi’ et al. in *Physica D* 240 (2011) 249 is not mentioned at all. In my opinion it should be at least commented for completeness.

Furthermore, it is not clear reading the paper how they conclude that the trapped particles do not move along the vortex core, but they are trapped and fixed on the vortex. Could they clarify this point in the paper?

There is also a mistake in the reference 24: the surname is “La Mantia” and the name is “Marco”. So it would be: La Mantia, M...

In conclusion the results and the idea of the paper are very interesting because it answers to some questions regarding the friction forces between quasi-particles and quantized vortices and because it determines the best mathematical model to use for describing the motion of the vortices. For this reasons, I suggest the publication of the paper after the authors have replied to my queries.

Reviewer #2 (Remarks to the Author):

In this paper, the authors present 4 experimental cases dealing with the dynamics of decorated quantum vortex rings in superfluid helium-4 decorated by Di-deuterium frozen particles. The first 2 cases focus on the decay of the ring radius over time due to friction forces between the vortex core and the particles with the fluid. Both of these experimental events are compared with numerical simulations. The first one with the Schwartz model, another model called 2W and a third one called S2W, while in the second case, only 2W and S2W are considered. After an analysis that is difficult to follow because it has lots of parameters—due to the fact that measurements are 2D and the dynamics of the ring is 3D—whose explanations are scattered over the entire manuscript, the authors conclude without any error-bar on their measured observables that the S2W model is the best model. The title of the paper and its main result is based on this conclusion that is weakly proved and discussed in my opinion. On top of these, the authors report the case of a ring that dives toward the gravity direction and hypothesize that it is due to the buoyancy of the particles attached on it, thanks to a comparison with the S2W that gives a questionable result. To finish, the case of a vortex ring that manages to detach from its particles is nicely physically discussed and the detachment speed (measured with questionable accuracy) is compared without error-bar to an estimation that uses an expression of the trapping force undiscussed in the paper.

I think that the experimental data reported in this paper are scarce but extremely valuable. They extend the work conducted in [28] in 2009 by a few more events and a first attempt of comparison with modern models. Nevertheless, this paper cannot be published in its present version. Major revisions are needed. Here after, I'll detail my main concerns.

Before going into detailed discussions, I wanted to stress general comments:

—> The temperature effect are never discussed in the paper:

One can read the temperature of the flow only in the legend of a figure and in the methods. The reader has to assume that all experiments are conducted at $T=1.65\text{K}$ with infinite precision. This is quite important in all the cases developed here since all the parameters (mutual friction forces, densities, viscous drag,...) depends on T .

—> One of the fundamental differences between the 2W and S2W model is the temperature dependence of the friction coefficients on T . In no way, a conclusion as strong as the one claimed in this paper can be proved without addressing this and even more fundamentally without error bars on the measured observables.

—> The amount of data used to reach the following conclusion : “This work eliminates long-standing ambiguities about the dissipative force on vortices” consist of the analysis of less than 1000 images. In my opinion, this does not allow to develop a reliable enough conclusion. Even if I do appreciate the processing that could lead to a strong conclusion with more data.

—> The numerical simulation are described in a very limited text in the methods. They are not conducted by the original authors of the S2W model and the particles implementation (crucial in the analysis presented in the paper) is presented as an obvious task when the original authors did not dare to do it yet. No “calibration” is discussed, none of the interpolations scheme discussed in [14] are discussed here. Moreover, the reader has to guess that these simulation where done on a 64^3 grid (unsure of course) and the effects of the discretization (in space and time) never discussed.

—> One can also regret that the seminal work [28] is not considered in this paper.

Hereafter I will come back on the different parts of the paper: Introduction, 9P ring event, 2P ring event, flipping ring event, detachment ring event, “discussion” and methods.

Introduction:

In my opinion, this introduction is general and clear enough to introduce this study. The reader can be surprised by the question asked in the very last part about the “triple ring structure” that is not discussed further in the paper. The last 2 sentences are far too strong at this stage to stay in their present form.

Results:

Visualizing quantized vortex rings

—> The critic of the seminal work [28] are very strong... : They did not produce useful data...? What is the meaning of “useful” in this sentence? The authors criticize the fact that this paper dealt with few data (1 ring trajectory) which is true (that we can understand considering the difficulty of these experiments). What is troublesome is the fact that the present study does not increase these data by an order of magnitude... Even if the movies provided present ~10 events only 4 different cases are quantitatively studied. This is also true that the ring studied in [28] is heavily loaded with particles, but this is extensively discussed in the paper, and this analysis is totally absent from the present study.

—> The authors of the present study also claim to “control” the ring generation. This is here again a statement that is too strong. When watching the movies provided, the reader easily realizes that the different events discussed are part of a collection that has many uncontrolled parameters (background flow, particle density, what is the trigger of the ring generation,...). This is further proved by words written by the authors : “occasionally”, “luckily”,... This does not lower the importance of these data, but reveal that these experiments can rich a better maturity. Along the manuscript the reader can read that “several” events where considered, but never finds how many... Are the movies provided the entire set of data? If it is the case it has to be stated clearly.

—> At this stage, it could be helpful for a reader to understand the particle generation process. It is only described quickly in the methods by mentioning a “slow” injection process of a D2/He mixture. What is the meaning of “slow”? This is important to share these information in order to help the community to reproduce these experiments : define the “optimal injection parameters”.

—> It is also intriguing to describe the particle trapping mechanism by invoking the pressure drop in the vortex core when considering D2 particles. Indeed, D2 particles are more dense (value not reported in the paper) than helium II, therefore the fictitious centrifugal force is stronger than the sum of the pressure forces applied by the surrounding fluid on the particle (Archimede): D2 particle should be expelled from the vortices... This trapping mechanism should be properly discussed in the manuscript.

Data analysis

—> The authors claim that “trapped particles do not move along the vortex core” but this statement (important in the process presented later) is not demonstrated in the paper. This has to be implemented.

General remark on the data

The analysis used to recover the particle sizes is very interesting but it is difficult to judge its accuracy when reading the paper. The particle sizes are recovered without any error bars based on a evolved algorithm that has the intensity of the image of the particles on the camera as main ingredient.

—> When watching the movies, one can see a lot of flickering and other background intensity variations. Additionally, particles used in the analysis are sometimes out of focus and looks like rings on the camera image. These experimental facts are not discussed, and surely affect the data. For example, the 9 particles are not visible in figure 1c, but traceable on the movie. What type of interpolation was used ? Effect on the data? Effect on the conclusions?

—> $B^*=1.17$ is a bit surprising. One could have expected $B^*=1$. What is the sensitivity of the conclusion to this value?

9P ring event (Fig 1)

—> The ellipse fit should provide an error bar on $R(t)$ that will be useful for model comparison. Models could also be modulated by the combined error of the particles diameters found through their intensity.

—> When $R(t)$ approach $200\ \mu\text{m}$, the trajectory of the ring seems to be perturbed in the movie. This doesn't reflect on Fig1e.

—> The comments on the efficiency of S2W to describe the data are inconsistent in this part of the paper. A new moderated conclusion could help the reader to follow the global argumentation.

—> The conclusion that S2W model is the better one in this case has to be discussed with error bars, since the best fit to the reported data seems to be S2W WITHOUT particles when the authors claim that in this 9P case, the number of particles on the ring is a strong problem. The difference with the 2W model loaded with particles is very tight... How much bigger needs to be the particles in the 2W model to fit the data? 1%, 10%, 100%?

2P ring event (Fig 2)

—> In this part, the uncertainty on the lightly decorated ring dynamic is very strong. All the constraints considered by the authors are reasonable but their accuracy is not. It seems to me that a self-consistency test is doable and would enrich the discussion. One could compare the intensity temporal evolution of both particle if the trajectory of the 2P were to be the one of the best fit produced by S2W model. According to what I can understand from the paper, these intensity traces will be almost constant. A fair comparison with the data would help the reader to evaluate the accuracy of the comparison.

—> The legend of the figure should be corrected to describe properly $I(t)$, which is a corrected $Im(t)$.

—> Fig2b is difficult to read.

—> c_1 and c_2 are claimed to be almost constant, can this statement be more quantitative? See their evolution, quantify their variations.

—> The lower value of $R(0)=d_p/2$ for all models right? (This could be clearly stated in the legend)

—> Is the upper value of $R(0)$ the same for all models?(This could be clearly stated in the legend)

—> Why not considering the Schwarz model in this section? There has to be a strong argument, because otherwise it is not obvious to me that the data cannot be compatible with it...

—> The fact that the best fit is $R(0)=d_p/2$ and that the ring has its velocity in the measurement plane is very lucky... The authors mention “other vortex ring events” that conduct to the same conclusion. How many? Are they all as luckily oriented and decorated ? Details are really needed.

Other intriguing observations

Flipping ring event

This movie like all other presented in the paper is precious and very interesting for the community. Nevertheless, the comparison with a S2W simulation giving a quantitative result of 36 particles of $4.9\ \mu\text{m}$ (again without error bars) weakens the entire paper and especially its main conclusion. On the movie one clearly sees 10 particles and no more, probably of diameter $1\ \mu\text{m}$ like the others. Moreover $36 \times 5 \times 2 = 360\ \mu\text{m}$... That doesn't fit on the ring at the end on its evolution. Additionally it is hard to imagine that only one solution exists to this problem. What is the algorithm used to produce this comparison? Accuracy?

Why not continue with the idea of comparing models and see if one of the 2 others produces a comparable trajectory with other N and diameters that could be a better match?

This part of the paper is for me very weak...

Detachment ring event

—>The physical discussion in this short paragraph is interesting. To be complete a short description of the expression of the maximum trapping force is expected (considering D_2 particles).

—>The perfect match between the detachment velocity and the measured observable should be discussed with error bars.

—> This last point raise the question of the velocity estimate in the entire paper. It has never been discussed. One has to assume that it is done using finite differences of particle position. This way of computing the velocity is heavily subject to the noise on the particle position. This should be discussed over the entire paper since it is the main observable used to compare with the models.

—> How do you deal with missing particles on a given image (because of flickering for example)? Interpolation scheme?

Discussion

—> The first sentence of this paragraph is not a discussion but an affirmation that is weakly supported by the manuscript as it is today. This should be corrected and extended.

—> Both of the extensions envisioned in the rest of the paragraph are interesting, but in both cases, vortex reconnexion will play a key role, and they will have to be implement ad hoc in the S2W model. This will need (like in the particles implementation) a tough work that need “calibration” and benchmarking.

Methods

Numerical models

I already commented in the first part of this report about these numerical models, here are some more detailed questions:

—> Resolutions (spatial and temporal): Where is the proof that the results presented here do not depend on these parameters?

—> How did you chose the values used in this study?

—> The scale separation between the resolution of the simulation and the inter-vortex spacing (R) becomes very weak at the end of the ring evolution (when the differences between models is the biggest). This has to be discussed.

Particle integration

—> The Magnus and Fns forces are considered as unchanged by the particle presence. What is the physical meaning of that?

—> The authors only considers Stokes drag, and buoyancy using [34] as a justification. But [34] deals with counterflow not a vortex ring. What is the argument to cancel the added mass force, the Basset history force,...? The point particle model can be a place to start, but its limits needs to be discussed. In particular, finite size effect, completely neglected here, should be discussed when $R \rightarrow 0$.

—> It is also mentioned that this model validity is subject to a particle separation much bigger than the particle diameter. This condition is not verified when $R \rightarrow 0$. A discussion is needed.

—> Orders of magnitude of the forces are evaluated close to $t=0$, where the velocity is the smallest and the differences between the models also. What happens when $t \rightarrow t_{collapse}$? Or event along the entire evolution?

—> General question: If one considers a vortex ring of radius R_0 with an initial velocity in the direction of the gravity decorated by 2 identical particles diametrically opposed, what is the critical Stokes number (vary size or density) above which the ring will deform out of a plane perpendicular to g using this model?

Particle size distribution:

—> Is the video with particle settling in a quiescent flow the only one used to build the correlation between intensity and size? This answer needs to be in the methods.

—> How did you deal with the flickering in this movie? Effect on the conclusions?

—> In the particle tracking algorithm, how do you define a particle? This is crucial when computing l_{mes} and worrisome when out of focus particles (like in 9P case) are considered despite their annulus shape.

—> Can you test your particle size measurement with the event of cluster formation from a ring? The volume of the particles should be conserved giving a relation between the particles present on the ring before the cluster and the cluster size. The verification of this relation by the measurement of the intensity and your algorithm will be a strong argument to prove that your procedure is correct.

I would like to finish this review by stressing the importance of these experimental data. They are rare and precious in this community, where experiments are extremely hard. This experiment, despite a certain lack of control, is precious and the data reported here also. Nevertheless, the conclusions of the present manuscript are too strong even if the results presented give a valuable hint. In the same spirit, the numerical simulations presented as a test case are very interesting but maybe not mature enough at this stage and deserve a serious benchmark in particular when considering the particle implementation (never done before on this model). I cannot accept this paper for publication as it is now, but I have no doubt that these authors can make it evolve to a very high quality scientific paper.

Reviewer #3 (Remarks to the Author):

The manuscript describes experimental observations of real-time trajectories of vortex loops in superfluid helium, and then compares them with numerical predictions of several two-fluid hydrodynamic models of various levels of complexity. At the end, the model with the most adequate description of the normal component's dynamics is shown to be the most consistent with all observations. While not very surprising in itself, this result is the first, and hence extremely important, experimental proof of the crucial role in quantum turbulence of the velocity field of the normal component – as was recently proposed theoretically. The whole manuscript (both its experimental and numerical parts of the work) is presented very well. I would hence recommend it for publication in Nature Communication subject to the following optional minor improvements:

1. It would be good to add a brief introduction into the microscopic origin of mutual friction, with references to its theoretical description and experimental measurements.
2. I would expect that, at certain low temperatures, the roton mean free path exceeds the relevant length scale of vortex loops, causing the hydrodynamics description of the normal component to break down. It would hence be useful if the authors comment on the temperature range of the applicability of their models, and whether the experimental 1.65 K is within it.
3. A vortex loop, especially immediately after being created by a self-reconnection, and under an external force from trapped particles, is generally not circular. However, in the manuscript, the shape of vortex loops is assumed to be that of a flat circular ring. This might be an appropriate approximation, but its validity needs to be discussed.
4. The reasons why trapped particles do not diffuse along the vortex core could have been discussed in more detail. The authors only write: "Interestingly, we find that the trapped particles do not move along the vortex core, which may support the core-damping idea proposed by Skoblin et al." (by the way, I personally do not quite understand this paper).
 - (i) As the Stokes mobility of these particles is known, it would be good to provide an estimate of the time required for a particle to travel some observable distance under the pull of the tangential component of the external (gravity plus buoyancy) force – and compare it with the time during which a ring was usually observed.
 - (ii) The normal component of the external force would create a sharp cusp at the location of the trapped particle. This is similar to the deformation of a vortex loop near a trapped electric charge in the presence of electric field. I would speculate that the presence of those cusps might greatly suppress the mobility of the trapped particles. The authors might wish to comment on this as well.

Dear Editor of *Nature Communications*,

We would like to express our gratitude to you and the reviewers for taking the time to review our manuscript. The thoughtful comments and feedback provided by the reviewers have been invaluable in improving the quality of our work. We have carefully considered the comments from all three reviewers and have addressed them in a point-by-point manner in the revised manuscript. Given the length of the comments provided by Reviewer 2, in what follows we present our responses to Reviewers 1 and 3 first, followed by our responses to Reviewer 2.

Report of Reviewer 1 and our responses:

The authors of the paper "Imaging quantized vortex rings in superfluid helium to decipher quantum dissipation" present some experimental results on superfluid helium by using visualization techniques. They use solidified deuterium tracer particles for decorating quantized vortices in order to study the dynamics of vortex rings and hence to analyze the friction forces between quantized vortices and the quasiparticles composing the normal fluid component in the two-fluid model.

In their studies they are in particular interested to compare their results with the theoretical ones coming from three existing mathematical models, which describe the motion of the quantized vortices in superfluid helium taking into account the back-reaction of the quasi-particles (normal component) to the vortices through the mutual friction force. According to their studies they conclude that the best model is that proposed by Galantucci et al. in *Eur. Phys. J. Plus* 135, 547 (2020).

It is surprising to me that the mathematical model proposed by Mongiovi' et al. in *Physica D* 240 (2011) is not mentioned at all. In my opinion it should be at least commented for completeness. Furthermore, it is not clear reading the paper how they conclude that the trapped particles do not move along the vortex core, but they are trapped and fixed on the vortex. Could they clarify this point in the paper?

Response: We are grateful to the reviewer for bringing to our attention the work by Jou, *et al.* Upon thorough examination of their paper, we have found that it focuses on a topic that differs significantly from the subject matter of our current work. Our primary concern is to identify the theoretical model that best explains the microscopic mutual friction acting on individual quantized vortex lines. We have evaluated three key models, namely the Schwarz model, the 2W model, and the S2W model, and have compared the results of the model simulations with our experimental data. On the other hand, Jou *et al.* discussed the generalization of the HVBK equations to describe the dynamics of the two fluid components in He II. It should be noted that the HVBK equations are applicable only at length scales greater than the intervortex distance l . These equations include a macroscopic mutual-friction force density term that should be calculated by integrating the microscopic mutual friction on individual vortex lines within a fluid parcel (with parcel size greater than l) divided by the parcel volume. The expression for this macroscopic mutual friction term in the existing literature is not suitable when the vortex tangle is anisotropic, polarized, and inhomogeneous. Jou *et al.* begin with the Schwarz description of the microscopic mutual friction on individual vortices (i.e., Eq. 3.1 and 3.4 in their paper) and systematically derive the macroscopic mutual friction term suitable for the HVBK model. Although their work is valuable, its focus is entirely different from that of our current paper. Nonetheless, we have cited their reference in our paper as one of the existing works that employ the Schwarz model.

For a vortex ring that carries more than 5 trapped particles, we can determine the precise locations of the particles on the ring and track their angular positions over time. We found that the particles do not exhibit significant displacements along the ring. Relevant details are given in our response to Reviewer-3 on page 4. We have also included an additional figure and discussions in the Methods section.

There is also a mistake in the reference 24: the surname is "La Mantia" and the name is "Marco". So it would be: La Mantia, M...

Response: We have corrected the name mistake in the reference as pointed out by the reviewer.

In conclusion the results and the idea of the paper are very interesting because it answers to some questions regarding the friction forces between quasi-particles and quantized vortices and because it determines the best mathematical model to use for describing the motion of the vortices. For this reasons, I suggest the publication of the paper after the authors have replied to my queries.

Response: We sincerely appreciate the reviewer for explicitly recommending our paper for publication in Nature Communications.

Report of Reviewer 3 and our responses:

The manuscript describes experimental observations of real-time trajectories of vortex loops in superfluid helium, and then compares them with numerical predictions of several two-fluid hydrodynamic models of various levels of complexity. At the end, the model with the most adequate description of the normal component's dynamics is shown to be the most consistent with all observations. While not very surprising in itself, this result is the first, and hence extremely important, experimental proof of the crucial role in quantum turbulence of the velocity field of the normal component – as was recently proposed theoretically. The whole manuscript (both its experimental and numerical parts of the work) is presented very well. I would hence recommend it for publication in Nature Communication subject to the following optional minor improvements:

Response: We thank the reviewer for explicitly recommending our paper for publication. In the following section, we provide responses to the questions raised by the reviewer.

1. It would be good to add a brief introduction into the microscopic origin of mutual friction, with references to its theoretical description and experimental measurements.

Response: Due to the word limit, we have added a concise sentence in the first paragraph that reads "As the vortices move through the normal fluid, a mutual friction between the two fluids can arise due to the scattering of the thermal quasiparticles off the vortex cores~\cite{Vinen-1957-PRS-I,Vinen-1957-PRS-III,Barenghi-1983-JLTP}".

2. I would expect that, at certain low temperatures, the roton mean free path exceeds the relevant length scale of vortex loops, causing the hydrodynamics description of the normal component to break down. It would hence be useful if the authors comment on the temperature range of the applicability of their models, and whether the experimental 1.65 K is within it.

Response: The roton mean free path is only about 30 nm at 1.0 K (see Fig.2 in PRB 75, 054506 (2007)) and it decreases with increasing the temperature. Therefore, the hydrodynamics description of the normal fluid in our experiment is completely reasonable. We have added a paragraph in the "Numerical models" subsection in Methods to make this point clear to the readers.

3. A vortex loop, especially immediately after being created by a self-reconnection, and under an external force from trapped particles, is generally not circular. However, in the manuscript, the shape of vortex loops is assumed to be that of a flat circular ring. This might be an appropriate approximation, but its validity needs to be discussed.

Response: When a vortex loop is created, it may not have a perfectly circular shape and can be deformed. These non-circular loops naturally vibrate due to their self-induced motion, which can be clearly seen in the first two ring events in Supplementary Video-2. However, mutual friction can effectively dampen these deformations and the associated vibrations. When analyzing the data, we only consider the vortex rings propagating smoothly in quiescent He II far from their creation moment, like the events presented in Fig. 1 and Fig. 2 of the paper. The smooth curves of the extracted ring radii over time in these figures indicate that these vortex rings do not vibrate and hence are essentially planar/circular.

4. The reasons why trapped particles do not diffuse along the vortex core could have been discussed in more detail. The authors only write: "Interestingly, we find that the trapped particles do not move along the

vortex core, which may support the core-damping idea proposed by Skoblin et al.” (by the way, I personally do not quite understand this paper).

Response: For a vortex ring with 5 or more trapped particles, we can determine the precise locations of the particles on the ring and track their angular positions over time. We found that the particles do not exhibit significant displacements along the ring. As an example, we show in the figure on the right the angular positions over time for a few particles trapped on the 9-particle ring presented in Fig. 1. This figure is now included in Methods for clarity. Skoblin *et al.* proposed a possible explanation for this phenomenon, but the scientific community has not yet reached a consensus on the underlying mechanism. We have revised the text to emphasize that this is still an open question and further research is needed to fully comprehend it.

(i) As the Stokes mobility of these particles is known, it would be good to provide an estimate of the time required for a particle to travel some observable distance under the pull of the tangential component of the external (gravity plus buoyancy) force – and compare it with the time during which a ring was usually observed.

Response: We have analyzed the total gravity force, which is $(\rho_p - \rho_{He}) \frac{4\pi}{3} a^3 g$, acting on the trapped particles for the 9-particle ring event shown in Fig. 1. The component of this force that acts tangentially to the vortex-ring perimeter varies depending on the location of the trapped particle and can reach up to about 40%. If we balance this force with the Stokes drag $6\pi a \mu_n u$, the corresponding terminal velocity u can exceed $50 \mu\text{m/s}$. Over an observational time of about 3 s, the trapped particle should move along the ring by about $150 \mu\text{m}$. However, the actual displacement we observed is an order of magnitude smaller. The nearly straight trajectories of the trapped particles shown in Fig. 1d also provide a clear indication that the particles do not move appreciably along the ring.

(ii) The normal component of the external force would create a sharp cusp at the location of the trapped particle. This is similar to the deformation of a vortex loop near a trapped electric charge in the presence of electric field. I would speculate that the presence of those cusps might greatly suppress the mobility of the trapped particles. The authors might wish to comment on this as well.

Response: In the work reported by Tsubota and Adachi (JLTP 158, 364-369 (2010)), an electric force of $1.6 \times 10^{-15} \text{ N}$ was applied to an electron bubble that was trapped on a vortex ring. This created a local cusp on the vortex ring with a size of approximately 1 nm. In our experiment, the net force of gravity and buoyancy acting on a typical trapped particle with a diameter of $2 \mu\text{m}$ was approximately $2.4 \times 10^{-15} \text{ N}$. Therefore, we expected a cusp of similar size to form. But this cusp size is much smaller than the spatial resolution of our visualization experiment and our vortex-filament simulation (i.e., $5 \mu\text{m}$), so we were unable to observe it. It might be possible that these cusps could assist in localizing the trapped particles as the reviewer postulated. But this hypothesis needs to be investigated in future numerical simulations, which is beyond the scope of our current study.

Report of Reviewer 2 and our responses:

In this paper, the authors present 4 experimental cases dealing with the dynamics of decorated quantum vortex rings in superfluid helium-4 decorated by Di-deuterium frozen particles. The first 2 cases focus on the decay of the ring radius over time due to friction forces between the vortex core and the particles with the fluid. Both of these experimental events are compared with numerical simulations. The first one with the Schwartz model, another model called 2W and a third one called S2W, while in the second case, only 2W and S2W are considered. After an analysis that is difficult to follow because it has lots of parameters - due to the fact that measurements are 2D and the dynamics of the ring is 3D - whose explanations are scattered over the entire manuscript, the authors conclude without any error-bar on their measured observables that the S2W model is the best model. The title of the paper and its main result is based on this conclusion that is weakly proved and discussed in my opinion. On top of these, the authors report the case of a ring that dives toward the gravity direction and hypothesize that it is due to the buoyancy of the particles attached on it, thanks to a comparison with the S2W that gives a questionable result. To finish, the case of a vortex ring that manages to detach from its particles is nicely physically discussed and the detachment speed (measured with questionable accuracy) is compared without error-bar to an estimation that uses an expression of the trapping force undiscussed in the paper.

Response: We thank the reviewer for providing us detailed comments. Our paper reports representative cases of four types of vortex-ring events: 1) the 9-particle ring case, presenting the events where the size and orientation of the vortex ring as well as the positions of the trapped particles can be completely determined; 2) the 2-particle ring case, representing the events where the particle effects minimally affect the ring's dynamics; 3) the flipping ring case, where particle's gravity controls the ring's dynamics; and 4) the clustering case, where trapped particles form a cluster as the ring shrinks. The first two cases were analyzed for the purpose of evaluating the accuracy of theoretical models on mutual friction, while the last two cases were included to report our novel observations and our interpretations. To enhance the reliability and persuasiveness of our analysis, we have added error bars to our data and made revisions throughout the paper. In what follows, we will address each question raised by reviewer.

I think that the experimental data reported in this paper are scarce but extremely valuable. They extend the work conducted in [28] in 2009 by a few more events and a first attempt of comparison with modern models. Nevertheless, this paper cannot be published in its present version. Major revisions are needed. Here after, I'll detail my main concerns.

Before going into detailed discussions, I wanted to stress general comments:

—> The temperature effect are never discussed in the paper:

One can read the temperature of the flow only in the legend of a figure and in the methods. The reader has to assume that all experiments are conducted at $T=1.65\text{K}$ with infinite precision. This is quite important in all the cases developed here since all the parameters (mutual friction forces, densities, viscous drag,...) depends on T .

Response: In our experiment, we control the temperature of the He II bath by regulating the vapor pressure using a computer-controlled solenoid valve connected to a pump. A feedback-based LabVIEW code was used to maintain the vapor pressure, ensuring that the temperature variation remains within 1 mK. This is a widely adopted method employed by many low-temperature research laboratories. We have added a brief description in the Results section to make this clear to the readers. We actually conducted experiments at a

few different temperatures. At higher temperatures, the lifetime of vortex rings is much shorter due to stronger dissipation. As a result, they can only move a short distance, making it much more challenging to observe them within our image plane. The number of events we obtained at higher temperatures was limited, and they were not of the same quality as those reported at 1.65 K. Nevertheless, we have included several vortex ring videos taken at higher temperatures in Supplementary Video-1 and explicitly explained the situation at the end of the first subsection in “Results”.

—> One of the fundamental differences between the 2W and S2W model is the temperature dependence of the friction coefficients on T. In no way, a conclusion as strong as the one claimed in this paper can be proved without addressing this and even more fundamentally without error bars on the measured observables.

Response: Temperature dependence is not the fundamental difference between the 2W model and the S2W model. The 2W model uses mutual friction coefficients obtained from second-sound attenuation measurements over an array of straight vortex lines in rotating helium II, and the mean normal-fluid velocity across the vortex array is used in the derivation. Conversely, the S2W model calculates the coefficients in a self-consistent manner by considering the local normal-fluid flow across each individual vortex line. More detailed discussions on the differences can be found in references [15-17]. To determine which model is more realistic, accurate data on the dynamical motion of the vortex ring must be provided and compared with the model's predictions. This is essentially what we are presenting in this paper. We have included error bars in our data as suggested by the reviewer and added relevant discussions in the paper.

—> The amount of data used to reach the following conclusion: “This work eliminates long-standing ambiguities about the dissipative force on vortices” consist of the analysis of less than 1000 images. In my opinion, this does not allow to develop a reliable enough conclusion. Even if I do appreciate the processing that could lead to a strong conclusion with more data.

Response: If the goal is to determine the value of a statistical quantity, then having more samples is certainly beneficial. However, the objective of the current research is to investigate the reliability of a dynamic model that describes the motion of a vortex ring. Therefore, it is crucial to obtain high-quality data that accurately captures the motion of the ring and provides quantitative information about the trapped particles for evaluating the particle effects. For this purpose, it is important to have images of rings that carry a suitable number of distinguishable particles so that the particle sizes and locations can be reliably determined. It is also important to have more images captured along the trajectory of the vortex ring per unit time, i.e., a high frame rate. Indeed, our frame rate of 200 Hz is among the highest of all existing vortex imaging experiments in He II.

—> The numerical simulation are described in a very limited text in the methods. They are not conducted by the original authors of the S2W model and the particles implementation (crucial in the analysis presented in the paper) is presented as an obvious task when the original authors did not dare to do it yet. No “calibration” is discussed, none of the interpolations scheme discussed in [14] are discussed here. Moreover, the reader has to guess that these simulation where done on a 64^3 grid (unsure of course) and the effects of the discretization (in space and time) never discussed.

Response: The numerical methods utilized in our research are all well-established, and their relevant details have been thoroughly reported in Ref. [17,18,23]. We do not understand why the reviewer criticized our

simulations “are not conducted by the original authors of the S2W model”, and we do not believe that “the original authors did not dare to” evaluate the particle effects on vortex-ring dynamics. In fact, we adopted the method reported in Ref. [39] to evaluate the particle effects, which was co-authored by some of our authors and the authors of the original S2W model papers. Furthermore, our S2W simulations were validated against the original work Ref. [17]. For instance, the normal-fluid velocity field shown in Fig. 1a using the S2W model agrees quantitatively with what was reported in Ref. [17]. The spatial and temporal resolutions of our 2W model simulations are provided in Methods, and these resolutions remain the same in the S2W model simulations. We have added a sentence to make it clear. Additionally, we have included information about the computational domain size, which was 120^3 grids. We have tried higher spatial resolution, i.e., 240^3 grids, and the results remain the same.

—> One can also regret that the seminal work [28] is not considered in this paper.

Response: Please see our detailed response at the bottom of this page.

Hereafter I will come back on the different parts of the paper: Introduction, 9P ring event, 2P ring event, flipping ring event, detachment ring event, “discussion” and methods.

Introduction:

In my opinion, this introduction is general and clear enough to introduce this study. The reader can be surprised by the question asked in the very last part about the “triple ring structure” that is not discussed further in the paper. The last 2 sentences are far too strong at this stage to stay in their present form.

Response: The triple-ring structure has been reported in multiple past numerical works, such as Ref. [15,17,24]. Considering the word limit of this article, we hope to direct readers to these references for further details instead of delving into the specifics of the triple-ring structure. Additionally, we have adjusted the last two sentences to reduce the strength of the statements.

Results:

Visualizing quantized vortex rings

—> The critic of the seminal work [28] are very strong... : They did not produce useful data...? What is the meaning of “useful” in this sentence? The authors criticize the fact that this paper dealt with few data (1 ring trajectory) which is true (that we can understand considering the difficulty of these experiments). What is troublesome is the fact that the present study does not increase these data by an order of magnitude... Even if the movies provided present ~10 events only 4 different cases are quantitatively studied. This is also true that the ring studied in [28] is heavily loaded with particles, but this is extensively discussed in the paper, and this analysis is totally absent from the present study.

Response: The work by Bewley and Sreenivasan (now Ref. [32]) was mentioned multiple times by the reviewer. We apologize if our original description caused any offense or gave the impression that we were criticizing this work, as that was not our intention. Ref. [32] is indeed a pioneering work on vortex ring imaging in He II. But the authors themselves noted that there were many tracers condensed on the core of the vortex ring, which could alter the core size and hence the ring’s dynamics. Because of the uncertainty surrounding the core size and the number and sizes of the trapped particles, it was not practical to conduct model simulations like what we presented in the current paper to determine which model is more reliable. Our original use of the word “useful” referred specifically to this point. We have revised the text to ensure

that readers are not left with the impression that we are criticizing Ref. [32].

Regarding the number of observed ring events, as we explained earlier, the most important factor is the quality of the data. Specifically, it is crucial that the data provide accurate information on the position and size of the ring, as well as the size and location of the trapped particles, in order to perform a quantitative calculation of the ring's motion using the numerical models. Ref. [32] reported only one ring event, which was likely due to the low vortex-line density in that experiment, making the creation of vortex rings through vortex reconnections unlikely. We refer to Ref. [32] to illustrate that we have made improvements to the experimental setup based on the insights gained from this pioneering work, including 1) the ability to control the line density by towing a grid through He II, and 2) the optimization of injection parameters to achieve a suitable particle number density along the vortex lines. These improvements have allowed us to obtain over two dozens of vortex ring events so that we can select a few events where individual trapped particles can be identified and the ring's dynamics can be calculated reliably.

—> The authors of the present study also claim to “control” the ring generation. This is here again a statement that is too strong. When watching the movies provided, the reader easily realizes that the different events discussed are part of a collection that has many uncontrolled parameters (background flow, particle density, what is the trigger of the ring generation,...). This is further proved by words written by the authors : “occasionally”, “luckily”,... This does not lower the importance of these data, but reveal that these experiments can reach a better maturity. Along the manuscript the reader can read that “several” events were considered, but never finds how many... Are the movies provided the entire set of data? If it is the case it has to be stated clearly.

Response: We'd like to clarify that we never claimed to control the generation of vortex rings. In our paper, we stated that we can "control the vortex generation". This is achieved by adjusting the towing speed of the grid through the He II-filled channel. Our purpose was to create a tangle of vortices to begin with. The decay of this tangle would increase the likelihood of observing vortex rings, as the reconnections of vortex lines in the tangle can give rise to them. However, in most of the image frames, we only observe quantized vortex lines, and vortex rings are seen “occasionally”. Despite this, our method is considered effective for vortex-ring research, as we've captured more than two dozens of vortex ring events in our experiment. We do not have a control over the number of particles trapped in a specific vortex ring, and the chances of observing vortex rings that only carry two particles are very slim. Hence, we considered it "lucky" to have observed such events. Due to the size limit, the uploaded videos only show representative ring events of the different types instead of all the captured ring events. We have revised the text in the first subsection in “Results” to make it clear to the readers.

—> At this stage, it could be helpful for a reader to understand the particle generation process. It is only described quickly in the methods by mentioning a “slow” injection process of a D2/He mixture. What is the meaning of “slow”? This is important to share these information in order to help the community to reproduce these experiments: define the “optimal injection parameters”.

Response: Our particle injection system is similar to the one described by Fonda *et al.* in Rev. Sci. Instrum. 87, 025106 (2016). The system involves passing a D2/He mixture through a needle valve and injecting it directly into He II. Due to the presence of the needle valve, the mass flow rate is quite low. Additional information about our injection system can be found in Rev. Sci. Instrum. 89, 015107 (2018) (Ref. [33]).

To ensure an appropriate number density of particles is trapped on vortices for vortex imaging, we conducted a substantial number of tests over the past a few years to determine the optimal injection conditions. These conditions include the injection pressure, mixture volume, injection duration, injection repetition rate, needle valve opening turns, and more. However, reporting the exact values of these parameters would be meaningless, as they are dependent on various factors such as the exact geometry of the injection pipelines, the inner diameter of the pipelines, the material and length of the injection tube, immersion depth of the injection tube, and so on. Thus, unless a researcher replicates our injection system and cryostat in precisely the same way, the optimal injection conditions may vary and need to be figured out through their own systematic tests.

—> It is also intriguing to describe the particle trapping mechanism by invoking the pressure drop in the vortex core when considering D2 particles. Indeed, D2 particles are more dense (value not reported in the paper) than helium II, therefore the fictitious centrifugal force is stronger than the sum of the pressure forces applied by the surrounding fluid on the particle (Archimede): D2 particle should be expelled from the vortices... This trapping mechanism should be properly discussed in the manuscript.

Response: We feel that the referee may have confused the particle trapping mechanism in classical viscous fluids with that in a superfluid. In a classical fluid, a particle located near a vortex tube is entrained by the viscous fluid and moves around the tube. Whether the particle moves towards or away from the tube then depends on the ratio of the particle density to the fluid density. However, in superfluid helium, a particle near a quantized vortex line is not entrained by the rotating inviscid superfluid and therefore does not experience the centrifugal force. Instead, the superfluid velocity increases as one moves closer to the vortex core. This increase in velocity results in a lower pressure on the side of the particle facing the vortex core, creating a net pressure that pushes the particle towards the vortex core. This trapping force acts on the particle regardless of its density and is essentially due to the Bernoulli effect, which is discussed in detail in Donnelly's book (Ref. [36]). The density of the D₂ particles is 202.8 g/cm³ as reported in Ref. [35]. We have included this information in the first subsection in "Results".

Data analysis

—> The authors claim that "trapped particles do not move along the vortex core" but this statement (important in the process presented later) is not demonstrated in the paper. This has to be implemented.

Response: For vortex rings with 5 or more trapped particles, we can determine the precise locations of the trapped particles on the ring and monitor their angular positions over time. We found that the particles do not display noticeable displacements along the ring. More details can be found in the figure and our response to the last question posted by the Reviewer-3. We have also included this figure and relevant discussion in the "Methods" section.

General remark on the data

The analysis used to recover the particle sizes is very interesting but it is difficult to judge its accuracy when reading the paper. The particle sizes are recovered without any error bars based on a evolved algorithm that has the intensity of the image of the particles on the camera as main ingredient.

Response: We have assessed the size uncertainty of individual particles by analyzing their brightness root mean variance ΔI during the time interval in which they were clearly detected by our tracking algorithm. We then calculate the upper and lower limits of the particle radius based on $\bar{I} + \Delta I$ and $\bar{I} - \Delta I$. The resulted

variation of the particle radius is now included in Extended Data Table 1 and has been incorporated into our revised model simulations. We have also added relevant explanations in the Methods section. It is worthwhile noting that since \bar{I} is nearly proportional to the square of the particle radius a^2 , an increase of \bar{I} by a factor of 2 (very rare) would only lead to 40% increase in a .

—> When watching the movies, one can see a lot of flickering and other background intensity variations. Additionally, particles used in the analysis are sometimes out of focus and looks like rings on the camera image. These experimental facts are not discussed, and surely affect the data. For example, the 9 particles are not visible in figure 1c, but traceable on the movie. What type of interpolation was used? Effect on the data? Effect on the conclusions?

Response: The brightness of a particle can be affected by many factors, such as its position within the laser sheet, its orientation (if the particle is not perfectly spherical), and the scattered light from nearby large particles. Considering these factors, we analyze all image frames where the particle can be clearly identified to calculate its mean brightness. We then use this mean brightness to determine the particle's radius by applying the radius correlation. For the 9-particle ring event shown in Fig. 1, some particles may become out of focus and thus invisible as the ring moves through the laser sheet. Nevertheless, we analyze the frames where these particles are clearly visible to determine their sizes and positions, which we then employ in the model simulation as initial conditions. We would also like to emphasize that the particle effects in the events we selected to present in this paper are quite small. For instance, even for the 9-particle event, the total drag force and gravity on the particles only account for 10%-18% and 4%-4.7% of the mutual friction acting on the bare ring as the ring shrinks from $R=310 \mu\text{m}$ to $150 \mu\text{m}$. In the revised manuscript, we have analyzed the particle size uncertainty and have incorporated this information in all the model simulations. It is clear that the particle-size uncertainty only has a minor effect on the model prediction.

—> $B^*=1.17$ is a bit surprising. One could have expected $B^*=1$. What is the sensitivity of the conclusion to this value?

Response: In an ideal scenario where particles have a spherical shape and light scattering is proportional to their surface area, the expected value of B^* should be 1. However, our best fit value of B^* is slightly larger, i.e., 1.17, which may be attributed to the particles' non-perfect shapes and the possibility that light penetrates slightly into the solid D2, causing light scattering not just on the particle surface but also from the particle interior. To establish the radius-brightness correlation, one could assume $B^*=1$ and alter the other fitting parameter. However, the resulting change in particle size is negligible, much smaller than that due to the brightness variance ΔI of the particles.

9P ring event (Fig 1)

—> The ellipse fit should provide an error bar on $R(t)$ that will be useful for model comparison. Models could also be modulated by the combined error of the particles diameters found through their intensity.

Response: We have included error bars for the $R(t)$ data in both Fig. 1 and Fig. 2. In Fig. 1, the error bars indicate the uncertainty of the fitting parameters resulting from the position uncertainties of the trapped particles in the ellipse fit. In Fig. 2, the error bars are calculated directly from the position uncertainty of the two trapped particles. In addition, we have incorporated the particle size uncertainty into our model simulations, which now account for the variation range of the simulation curves when the particle radius changes from $a-\Delta a$ to $a+\Delta a$.

—> When $R(t)$ approach $200\ \mu\text{m}$, the trajectory of the ring seems to be perturbed in the movie. This doesn't reflect on Fig 1e.

Response: The reviewer referred to a perturbation that occurs at around 2 s in the original Supplementary Video-3 for the 9-particle ring event. This “perturbance” is actually a reconnection between the vortex ring and another vortex line, which results in the ring's destruction and marks the end of the data curve presented in Fig. 1e. The video only shows the ring's motion for about 2 s before the reconnection event, while Fig. 1e contains the $R(t)$ data for over 3 s. This is because we did not upload the full video of the 9-particle ring due to the size limit of the uploaded files. To avoid any confusion, we have replaced the original Video-3 with the full version one, which aligns the video time with the time in Fig. 1e.

—> The comments on the efficiency of S2W to describe the data are inconsistent in this part of the paper. A new moderated conclusion could help the reader to follow the global argumentation.

—> The conclusion that S2W model is the better one in this case has to be discussed with error bars, since the best fit to the reported data seems to be S2W WITHOUT particles when the authors claim that in this 9P case, the number of particles on the ring is a strong problem. The difference with the 2W model loaded with particles is very tight... How much bigger needs to be the particles in the 2W model to fit the data? 1%, 10%, 100%?

Response: The first two paragraphs in the “Data analysis and model comparison” subsection are pertinent to the discussion of the 9-particle event. Towards the end of these two paragraphs, we concluded that: “Obviously, the Schwarz model over-estimates the dissipation and can be rejected. But it becomes less clear whether the S2W model still describes the data better than the 2W model. To make a reliable assessment on these two models, it is imperative to analyze rings with a minimal number of trapped particles, since the uncertainties in the particle sizes and positions could affect both the $R(t)$ data and the simulated curves.”

Clearly, our conclusion was not that the S2W model is superior to the 2W model based on the 9-particle ring analysis alone. Instead, we concluded that a further analysis of rings with fewer particles is required to make a reliable judgment between the S2W model and the 2W model. Later, in our analysis of the 2-particle ring, we demonstrated that the S2W model provides a better agreement with the observation than the 2W model. To this end, we do not see any inconsistent statements. Following the reviewer's suggestion, we have included error bars for the experimental data and evaluated the effect of particle size uncertainties on the simulation curves.

2P ring event (Fig 2)

—> In this part, the uncertainty on the lightly decorated ring dynamic is very strong. All the constraints considered by the authors are reasonable but their accuracy is not. It seems to me that a self-consistency test is doable and would enrich the discussion. One could compare the intensity temporal evolution of both particle if the trajectory of the 2P where to be the one of the best fit produced by S2W model. According to what I can understand from the paper, these intensity traces will be almost constant. A fair comparison with the data would help the reader to evaluate the accuracy of the comparison.

Response: Fig. 2c displays the temporal evolution of the corrected brightness of the two trapped particles. A mild decrease of the corrected brightness over time is observed, which is due to the particles' displacement perpendicular to the laser plane, as we described in the Methods section. We incorporated this

information in our model simulations to constrain the projection angle of the ring. Therefore, the model that gives the best fit to the data naturally should render particle-brightness curves in agreement with those shown in Fig. 2c. In the figure on the right, the simulated brightness curves using the best fitted S2W model are compared with the data, where a reasonable agreement is seen. We have included this figure in Fig. 2c.

—> The legend of the figure should be corrected to describe properly $I(t)$, which is a corrected $I_m(t)$.

Response: We have updated the figure caption to clarify that $I(t)$ denotes the corrected brightness.

—> Fig2b is difficult to read.

Response: We have enlarged Fig. 2b as much as we can and adjusted the positions of the parameter labels.

—> c_1 and c_2 are claimed to be almost constant, can this statement be more quantitative? See their evolution, quantify their variations.

Response: c_1 is the ratio of the initial separation distance $d(0)$ between the two trapped particles to the initial ring radius $R(0)$ assumed in the model simulation. This ratio would remain constant as long as the particles do not move along the vortex core, which has been supported by the analysis of the rings with more than 5 particles where the particles' angular position can be determined. c_2 is Cosine of the angle between the ring's propagation direction and the laser plane. It remains constant as long as the ring's trajectory is straight, i.e., unaffected by particles' gravity or other forces. We have monitored the center of the two particles and confirmed that its trajectory is straight. The straight trajectory can also be inferred from Fig. 2a.

—> The lower value of $R(0) = d_p/2$ for all models right? (This could be clearly stated in the legend)

—> Is the upper value of $R(0)$ the same for all models?(This could be clearly stated in the legend)

Response: As we have explained in the text, the lower limit of $R(0)$ is set by the combined constraints of $R(0) \geq d_p/2$ and $u(0) \leq u_p(0)/c_2$, while the upper limit of $R(0)$ is set by $u(0) \geq u_p(0)$. Note that, different models give different ring velocities even with the same ring radius. Therefore, the velocity constraints can lead to different allowable range of $R(0)$ for the different models. We have revised the figure caption to refer to this information.

—> Why not considering the Schwarz model in this section? There has to be a strong argument, because otherwise it is not obvious to me that the data cannot be compatible with it...

Response: Based on the analysis of the 9-particle ring event, we can confidently reject the Schwarz model. This is evident in Fig. 1e, where the simulation curve of the Schwarz model (with particles) is far off from the experimental data. However, it remained uncertain whether the 2W model or the S2W model was a more accurate description of the 9p ring data. For this reason, we presented additional analysis of the 2-particle ring event in the paper to specifically assess the 2W and the S2W models. We have explained this

reasoning in the paper. Furthermore, we have also conducted the Schwarz model calculation for the 2p ring event. As shown in the figure below, the tuning range of the Schwarz model is again off from the data.

—> The fact that the best fit is $R(0)=d_p/2$ and that the ring has its velocity in the measurement plane is very lucky... The authors mention “other vortex ring events” that conduct to the same conclusion. How many? Are they all as luckily oriented and decorated? Details are really needed.

Response: The analysis of the two-particle ring event involves the c_1 parameter, which is the ratio of the separation between the two particles and the diameter of the ring. This approach is applicable to all general cases, regardless of whether the two trapped particles are located across the ring’s diameter or not. In Fig. 2, we presented our best event where the background flow was negligible and there were no large particles moving nearby when the ring was recorded. Although the two particles happen to be located nearly across the diameter, this does not imply an advantage or ease in the analysis. We have collected five two-particle vortex-ring events, and in some cases, the fitted ring diameter was larger than the particle separation. Due to the page limit set by the journal, we cannot present the analysis of all the ring events. However, we plan to report more detailed information in a specialized journal in the future.

Other intriguing observations

Flipping ring event

This movie like all other presented in the paper is precious and very interesting for the community. Nevertheless, the comparison with a S2W simulation giving a quantitative result of 36 particles of $4.9\ \mu\text{m}$ (again without error bars) weakens the entire paper and especially its main conclusion. On the movie one clearly sees 10 particles and no more, probably of diameter $1\ \mu\text{m}$ like the others. Moreover $36 \times 5 \times 2 = 360\ \mu\text{m}$... That doesn’t fit on the ring at the end on its evolution. Additionally it is hard to imagine that only one solution exists to this problem. What is the algorithm used to produce this comparison? Accuracy? Why not continue with the idea of comparing models and see if one of the 2 others produces a comparable trajectory with other N and diameters that could be a better match?

This part of the paper is for me very weak...

Response: We feel that the reviewer may have misunderstood the purpose of the section on the flipping rings. We present the flipping ring event (and the clustering event) **NOT** for the purpose of comparing with model simulations and to assess the reliability of the models. The assessment on different models was done through the analysis of the 9P ring and the 2P ring events, where the particle effects can be quantitatively analyzed. As we stated at the beginning of this section, our primary goal was to present some novel

phenomena that have not been previously reported in the literature. All the observed flipping rings contain many particles, and as they shrink, the trapped particles can aggregate and form solid rods along the vortex core, similar to the situation reported in Ref. [32]. Counting the exact number of trapped particles and evaluating their positions and sizes is impractical. We conducted the S2W model simulation solely for the purpose of illustrating qualitatively our understanding that the ring can flip downward due to the gravity effect if there are enough trapped particles with large sizes. In principle, any model can be used to demonstrate this concept. But since we concluded in the earlier section that the S2W model provides a better description of the observed ring's motion, naturally we would prefer to use it. We would also like to clarify that the solution is not unique, and similar flipping can be achieved with fewer but heavier trapped particles. We have added a description to make this point clear to the readers.

Detachment ring event

—>The physical discussion in this short paragraph is interesting. To be complete a short description of the expression of the maximum trapping force is expected (considering D2 particles).

Response: The binding energy of a particle near a vortex line can be estimated by integrating the kinetic energy of the superfluid originally occupying the particle's volume. This binding energy is a function of the distance between the particle center and the vortex core. By taking a derivative of the binding energy with respect to this distance, an expression for the trapping force can be obtained. This trapping force is negligible far from the vortex core, increases as the particle approaches the core, and eventually reaches zero again when the particle reaches the core center. Therefore, the force must be maximized at a finite distance from the core center. Meichle and Lathrop derived an approximate expression in Ref. [42], assuming the maximum trapping force occurs at a distance comparable to the particle radius. This trapping force expression is independent of the ratio of the particle density to the superfluid density. Considering the word limit of this paper, we must refer readers to Ref. [42] for further information.

—>The perfect match between the detachment velocity and the measured observable should be discussed with error bars.

Response: While the estimated detachment velocity closely matches with the observed one, this should not be overemphasized due to the rough and approximate nature of the expression for the trapping force, as explained in Ref. [42]. Nonetheless, we have included error bars in all the velocity data presented throughout the paper for clarity.

—> This last point raise the question of the velocity estimate in the entire paper. It has never been discussed. One has to assume that it is done using finite differences of particle position. This way of computing the velocity is heavily subject to the noise on the particle position. This should be discussed over the entire paper since it is the main observable used to compare with the models.

Response: In order to aid the discussion, we show in the following figure the time evolution of the x and z coordinates of the center of the vortex ring presented in Fig. 3c. We calculate the velocity at t via a linear fit to the position-versus-time data collected over 5 successive image frames spanning from $t-2\Delta t$ to $t+2\Delta t$. The fitted slope gives the velocity. This procedure is adopted in all of our velocity calculations. We have added error bars associated with the fit to all the velocity data and have added relevant discussions in “Methods”.

—> How do you deal with missing particles on a given image (because of flickering for example)? Interpolation scheme?

Response: A particle may become invisible if it moves out of the imaging plane. However, in the time period that the particle can be clearly observed, we can measure its position and size. This information is then used in our model simulation. Specifically, at $t=0$ in the simulation, we place all the detected particles on the ring based on their recorded positions. There is no interpolation implemented in our analysis. The brightness of a particle is analyzed for the entire period that it is detected. The mean brightness is then used to evaluate the particle size. Some additional information has been provided in our response to an earlier inquiry on page 10.

Discussion

—> The first sentence of this paragraph is not a discussion but an affirmation that is weakly supported by the manuscript as it is today. This should be corrected and extended.

Response: We have revised this sentence.

—> Both of the extensions envisioned in the rest of the paragraph are interesting, but in both cases, vortex reconnection will play a key role, and they will have to be implemented ad hoc in the S2W model. This will need (like in the particles implementation) a tough work that needs “calibration” and benchmarking.

Response: After a reconnection, the participating vortex lines exhibit high-speed motion through the normal fluid, and the different models may yield disparate outcomes. Our discussions emphasize the importance of employing the S2W model when studying the vortices in such high-speed motion. We would also like to mention that we have collected a large set of vortex reconnection data and are currently analyzing them to generate instructive information on the proper implementation of vortex reconnections in the vortex filament model. This work will be reported in a future publication.

Methods

Numerical models

I already commented in the first part of this report about these numerical models, here are some more detailed questions:

—> Resolutions (spatial and temporal): Where is the proof that the results presented here do not depend on these parameters?

—> How did you chose the values used in this study?

Response: As we responded earlier, the spatial and temporal resolutions of our 2W model simulations are provided in Methods, i.e., $\Delta x = \Delta y = \Delta z = 0.0083$ mm and $\Delta t = 10^{-5}$ s. These resolutions remain the same in the S2W model simulations. The computational domain consists of 120^3 grids. We have conducted simulations using finer resolutions (for instance 240^3 grids) and confirmed that the simulation results such as the $R(t)$ curves do not exhibit any discernible changes. Relevant discussions have been added in Methods. We chose the specified resolutions to strike a balance between achieving result convergence and ensuring efficient computation speed.

—> The scale separation between the resolution of the simulation and the inter-vortex spacing (R) becomes very weak at the end of the ring evolution (when the differences between models is the biggest). This has to be discussed.

Response: As specified in Methods, we discretize the vortex rings in our filament model simulations with a spatial resolution $\Delta \zeta = 5$ μm . When we make quantitative comparison between experimental data and the model simulations, we always focus on the regime where the ring's perimeter is far larger than both the size of the particles and $\Delta \zeta$. Specifically, the perimeter measures approximately 750 μm for the 9P ring event and around 150 μm for the 2P ring event at the tail end of their respective $R(t)$ curves. Thus, the issue of scale separation does not arise in our work.

Particle integration

—> The Magnus and Fns forces are considered as unchanged by the particle presence. What is the physical meaning of that?

Response: In Ref. [39], a detailed discussion is presented regarding the expressions of all the forces acting on a vortex segment that carries a small trapped particle, including the Magnus force \mathbf{f}_M and the mutual friction force \mathbf{f}_{ms} . When the size of the particle is small compared to the length of a vortex segment, the expressions for \mathbf{f}_M and \mathbf{f}_{ms} are identical to those for a bare vortex segment. Alternatively, one may consider the total force acting on the ring. For example, the total mutual friction force is proportional to the length of the ring's perimeter. Even for our 9P ring event, the trapped particles occupy less than 2% of the vortex core length over the entire propagation of the ring. As a result, the total mutual friction force is nearly equal to that of a bare ring of the same size.

—> The authors only considers Stokes drag, and buoyancy using [34] as a justification. But [34] deals with counterflow not a vortex ring. What is the argument to cancel the added mass force, the Basset history force,...? The point particle model can be a place to start, but its limits needs to be discussed. In particular, finite size effect, completely neglected here, should be discussed when $R \rightarrow 0$.

Response: We would recommend the reviewer to check the paper by Mineda *et al.* (now Ref. [39]) for a comprehensive understanding of this valuable work. These authors developed a general theoretical framework for assessing the impacts of particles on vortex dynamics. This framework is not only applicable to vortex-tangle dynamics in counterflow turbulence but also to the much simpler case of a single vortex ring. In our analysis, we incorporated the added mass effect (which is small). We also evaluated the Basset force caused by the acceleration of the ring, but we arrived at the same conclusion as in Ref. [39], namely

the acceleration force is negligible compared to other terms in the equation of motion. Moreover, we never analyzed any data where R is so small such that the particle-effect model of Ref. [39] becomes unreliable.

—> It is also mentioned that this model validity is subject to a particle separation much bigger than the particle diameter. This condition is not verified when $R \rightarrow 0$. A discussion is needed.

Response: We never analyze any data in the limit $R \rightarrow 0$ when assessing the different models. When we analyze the 9P ring event and the 2P ring event, the particle separation is always much larger than the particle size. If the reviewer was referring to the clustering event where the ring shrinks to zero size, we did not conduct any quantitative analysis using any of the models.

—> Orders of magnitude of the forces are evaluated close to $t=0$, where the velocity is the smallest and the differences between the models also. What happens when $t > t_{\text{collapse}}$? Or event along the entire evolution?

Response: In the Methods section, we presented the ratio of forces at $t=0$ to illustrate that the impact of particles on the 2P ring event is significantly smaller than that on the 9P ring event. In response to the reviewer's comment, we have added further details to clarify how the force ratios change as the ring shrinks. For the 9P ring event, the ratio of drag to F_{ns} increases from about 10% at $R=310 \mu\text{m}$ to about 18% at $R=150 \mu\text{m}$, while the ratio of gravity to F_{ns} only increases from 4% to 4.7%. In contrast, for the 2P ring event, despite its small radius and hence larger drag, drag/ F_{ns} only rises from approximately 4.8% at $R=140.8 \mu\text{m}$ to about 10% at $50 \mu\text{m}$, whereas gravity/ F_{ns} only increases from 0.8% to 0.83%.

—> General question: If one considers a vortex ring of radius R_0 with an initial velocity in the direction of the gravity decorated by 2 identical particles diametrically opposed, what is the critical Stokes number (vary size or density) above which the ring will deform out of a plane perpendicular to g using this model?

Response: Please refer to our response to the last question of reviewer-3. Although the gravity of the particle we used in our experiment can cause local deformation of the vortex line, this effect is so minimal that it cannot be observed in either our experiment or numerical simulation. One may conduct additional numerical simulations with gradually increasing particle size or density to figure out the threshold Stokes number above which the ring develops an appreciable local deformation. However, this type of simulation would require a significant amount of computation time and resources, which is beyond the scope of the current study. We may consider this topic in our future research.

Particle size distribution:

—> Is the video with particle settling in a quiescent flow the only one used to build the correlation between intensity and size? This answer needs to be in the methods.

Response: We have acquired video data showing the particle settling in each experimental run. The particle settling video that we have uploaded (i.e., Supplementary Video-7) was a part of the one taken in the experimental run where we recorded the 9P ring event and the 2P ring event and is therefore directly relevant to the presented data analysis. We have added a discussion in Methods to clarify this.

—> How did you deal with the flickering in this movie? Effect on the conclusions?

Response: For the videos showing the settling motion of the particles, we analyze all image frames to determine the settling velocity and brightness of each particle. We then generate the probability distributions for the settling velocity and the brightness. The size-brightness correlation is established by

comparing these distributions, as we discussed in detail in Methods. The flickering of the brightness would widen the brightness distribution. We do not conduct any special treatment of this effect. When we analyze the size of a trapped particle, we again analyze all image frames where the particle can be distinctly identified to calculate its mean brightness, where similar flickering effect still exists such that the analysis is fair. Note again the particle effects for the rings we selected to present are relatively small. In the updated Fig. 1 and Fig. 2, we have incorporated the particle size variations based on their brightness variation from $\bar{I} + \Delta I$ to $\bar{I} - \Delta I$. We observed only minor changes in the simulation curves.

—> In the particle tracking algorithm, how do you define a particle? This is crucial when computing I_{mes} and worrisome when out of focus particles (like in 9P case) are considered despite their annulus shape.

Response: As stated in the paper, we utilized a feature-point tracking routine (Ref. [37]) to identify particles, which essentially involves a Gaussian fit of the particle image profile to determine its center location. This approach can be applied even when the particle is slightly out of focus, with the Gaussian fit still providing the particle center position, albeit with larger associated uncertainty. We have included the particle location uncertainty in our analysis, which contributes to the error bars in the ring radius $R(t)$ curve in the updated figures.

—> Can you test your particle size measurement with the event of cluster formation from a ring? The volume of the particles should be conserved giving a relation between the particles present on the ring before the cluster and the cluster size. The verification of this relation by the measurement of the intensity and your algorithm will be a strong argument to prove that your procedure is correct.

Response: The clustering event shown in Fig. 3c involved five trapped particles, and their estimated radiuses are as follows: $1.21 \pm 0.07 \mu\text{m}$, $1.26 \pm 0.08 \mu\text{m}$, $1.36 \pm 0.12 \mu\text{m}$, $1.37 \pm 0.11 \mu\text{m}$, $1.63 \pm 0.19 \mu\text{m}$. Assuming that the total volume remains constant, the expected radius of the merged cluster would be $2.36 \pm 0.21 \mu\text{m}$. We have also estimated the merged cluster's radius based on its brightness, and the obtained cluster radius is $2.62 \pm 0.11 \mu\text{m}$. This result is only slightly larger than the expected value, which could be attributed to the possibility that the merged cluster may have voids in between the particles.

I would like to finish this review by stressing the importance of these experimental data. They are rare and precious in this community, where experiments are extremely hard. This experiment, despite a certain lack of control, is precious and the data reported here also. Nevertheless, the conclusions of the present manuscript are too strong even if the results presented give a valuable hint. In the same spirit, the numerical simulations presented as a test case are very interesting but maybe not mature enough at this stage and deserve a serious benchmark in particular when considering the particle implementation (never done before on this model). I cannot accept this paper for publication as it is now, but I have no doubt that these authors can make it evolve to a very high quality scientific paper.

REVIEWERS' COMMENTS

Reviewer #1 (Remarks to the Author):

The authors have replied to my questions, so I suggest the publication of the paper.

Reviewer #2 (Remarks to the Author):

The authors have considered all of our remarks and answered very clearly (in the main text or in their answer) all points addressed in the first round of reviews. As I pointed out in my first review, these data, simulations and analysis are very stimulating and important (as proved by the length of my first report). In my opinion, the revisions proposed by the authors qualify this paper for publication in Nature Comm.

I do recommend this paper for publication as it is now.

I can't wait to read the sequels they announced in their answer to the reviewers.

Reviewer #3 (Remarks to the Author):

The revised manuscript is greatly improved. The authors answered all my questions (as well as all questions of other referees) to my complete satisfaction. This is an excellent piece of cutting-edge science and, in my opinion, should be published in Nature Communications as it is now.

Florida State University
1800 East Paul Dirac Drive
Tallahassee, Florida 32310
nationalmaglab.org

Dear Editor of *Nature Communications*,

We would like to express our gratitude to you and the reviewers for taking the time to review our manuscript. All three reviewers have recommended the publication of our manuscript without any further revisions. There is no question or comment that we need to address.

Best regards,

Dr. Wei Guo on behalf of the authors